# Structuring Collective Action with LLM-Guided Evolution: From Ill-Structured Problems to Executable Heuristics

## Abstract

Collective action problems, which require aligning individual incentives with collective goals, are classic examples of Ill-Structured Problems (ISPs). For an individual agent, the causal links between local actions and global outcomes are unclear, stakeholder objectives often conflict, and no single, clear algorithm can bridge micro-level choices with macro-level welfare. We present ECHO-MIMIC, a general computational framework that converts this global complexity into a tractable, Well-Structured Problem (WSP) for each agent by discovering executable heuristics and persuasive rationales. The framework operates in two stages: ECHO (Evolutionary Crafting of Heuristics from Outcomes) evolves snippets of Python code that encode candidate behavioral policies, while MIMIC (Mechanism Inference & Messaging for Individual-to-Collective Alignment) evolves companion natural language messages that motivate agents to adopt those policies. Both phases employ a large-language-model-driven evolutionary search: the LLM proposes diverse and context-aware code or text variants, while population-level selection retains those that maximize collective performance in a simulated environment. We demonstrate this framework on two distinct ISPs: a canonical agricultural landscape management problem and a carbon-aware EV charging time slot usage problem. Results show that ECHO-MIMIC discovers high-performing heuristics compared to baselines and crafts tailored messages that successfully align simulated agent behavior with system-level goals. By coupling algorithmic rule discovery with tailored communication, ECHO-MIMIC transforms the cognitive burden of collective action into a implementable set of agent-level instructions, making previously ill-structured problems solvable in practice and opening a new path toward scalable, adaptive policy design.

## 1 Introduction

Many of the most pressing real-world challenges, from sustainable resource management and climate change mitigation to economic policy design, are Ill-Structured Problems (ISPs) (Simon & Newell, 1971; Reitman, 1964). Unlike Well-Structured Problems (WSPs), which have clearly defined goals, known constraints, and a finite set of operators, ISPs feature ambiguous goals, unclear causal relationships, and undefined solution spaces (Simon, 1973). Solving an ISP requires the problem-solver to impose structure, define objectives and discover pathways, as an integral part of the solution process itself.

A classic example of an ISP arises in collective action problems, where locally rational decisions made by autonomous agents lead to globally suboptimal or even harmful outcomes (Hardin, 1968; Ostrom, 1990). Consider farmers operating within a shared agricultural landscape or EV owners choosing when to charge at home. Each agent makes decisions driven by local incentives like maximizing crop yield or minimizing charging costs and discomfort. While these decisions may be optimal at the individual level, their combined effect can degrade the shared ecosystem or overload the grid during peak hours. For an individual agent, the decision of how to act is an ISP: the link between their specific choices and the health of the entire system is complex and unclear, and the *right* action is not algorithmically defined. The challenge for a system designer or policymaker is to create a mechanism that simplifies this decision-making process for the individual. An ideal

solution would be practical behavioral rules, or heuristics, that, if followed by individual agents, reliably produce a desirable global outcome. Such heuristics would effectively transform the ISP faced by each agent into tractable WSPs. Discovering such heuristics, however, is a challenging second-order problem.

We introduce **ECHO-MIMIC**, a framework designed to automate the discovery of these heuristics and the mechanisms to encourage their adoption. Our approach is grounded in Simon's models of bounded rationality, which posit that agents rely on rules-of-thumb to navigate complex environments (Gigerenzer & Gaissmaier, 2011; Simon, 1990). We operationalize this concept using the synergy of Evolutionary Algorithms (EAs) and Large Language Models (LLMs). This LLM+EA paradigm represents a new frontier for creative problem-solving, and recent work has begun to leverage LLMs within evolutionary program searches to generate and tune heuristics for complex optimization problems (Guo et al., 2023; Romera-Paredes et al., 2024; Liu et al., 2024a; Ye et al., 2024; Novikov et al., 2025; Chen et al., 2023). However, the utility of this paradigm in practical optimization settings and its applicability to real-world complex systems has been underexplored.

Our end-to-end framework leverages this paradigm to solve real-world collective action problems, transforming them from ISPs into effective WSPs. Our primary contributions are:

1. We introduce ECHO-MIMIC, a general framework that deconstructs complex collective action ISPs into executable behavioral heuristics that are well-structured for individual agents, and then nudges the agents to implement these heuristics.

2. We demonstrate our framework on two distinct domains: agricultural landscape management and carbon-aware EV charging. We show that it significantly outperforms LLM program-synthesis baselines like DSPy MiPROv2 and agent frameworks like AutoGen.

3. We find that performance of heuristics produced by ECHO rises with code-complexity indicators and that nudges generated by MIMIC can be tailored to diverse agent personas.

4. To facilitate generalization, we develop a Domain Creation Agent that automatically generates modular, domain-specific system instructions and prompts given the (state, action) schema and constraints of a new task.

5. Peripherally, we show the effectiveness of the LLM+EA paradigm on optimization problems in real-world systems, moving beyond work focusing on combinatorial benchmarks (Liu et al., 2024a; Ye et al., 2024; Dat et al., 2025; Romera-Paredes et al., 2024).

## 2 RELATED WORK

**LLM-guided evolutionary search and automated heuristic design:** A growing line of work couples LLMs with evolutionary search to generate programs, prompts, and heuristics. FunSearch demonstrates LLM-driven program discovery within an evolutionary loop for mathematical problems (Romera-Paredes et al., 2024). EvoPrompt connects LLMs with evolutionary algorithms to evolve high-performing prompts (Guo et al., 2023). LLMs have also been used as evolutionary optimizers or operators more broadly (Liu et al., 2024b; Yang et al., 2023; Lange et al., 2024). Beyond prompts, language hyper-heuristics (Burke et al., 2003) evolve executable code to improve search efficiency and generality across combinatorial problems (Ye et al., 2024; Liu et al., 2024a; Dat et al., 2025). Our ECHO phase aligns with this paradigm but specializes it to produce validated code heuristics that map local states to actions to drive collective-action.

**Multi-agent optimization and communication:** Recent frameworks like DSPy (Khattab et al., 2023; Opsahl-Ong et al., 2024) and AutoGen (Wu et al., 2024) enable the construction of multi-agent LLM systems for diverse tasks. DSPy provides a programming model for optimizing LM prompts and weights through compilation, and AutoGen provides a flexible infrastructure for agent interaction. While these frameworks are very useful in constructing multi-agent systems and workflows for general problems, they do not inherently solve the problem of discovering optimal local policies for collective goals in complex, constraint-heavy environments, but rather need to be explicitly setup to do so. G-Designer (Zhang et al., 2024) addresses the design of multi-agent communication topologies via graph neural networks, which is related but orthogonal to our work. Our setup fixes the neighbor graph and focuses on program (policy) synthesis + measurable nudging. We compare against AutoGen as a general-purpose agent scaffold baseline and DSPy MIPROv2 (Opsahl-Ong et al., 2024) as a strong prompt optimization baseline.

**AI and social dilemmas:** Within AI, a large body of work has studied social dilemmas in synthetic multi-agent substrates. Sequential and intertemporal social dilemmas in grid-world or DMLab-style environments have been used to analyze emergent cooperation under different learning rules and reward structures (Leibo et al., 2017; Peysakhovich & Lerer, 2017). Other work rewards social influence or inequity aversion to improve coordination (Jaques et al., 2019; Hughes et al., 2018). Learning-to-incentivize approaches similarly optimize incentive functions in simulated MARL tasks without targeting concrete deployments (Yang et al., 2020). Most of these benchmarks use stylized spatial geometries and focus on optimizing opaque neural policies or reshaped rewards. Our method is an end-to-end way to approach social dilemmas with deployable rule-books and nudging, and we demonstrate it in real-world settings. Moreover, our agents are bounded-rational rule users that employ executable heuristics, which is closer to how humans make decisions.

**Collective action, bounded rationality, and heuristics:** The core challenge we target, aligning individual incentives with social welfare, sits squarely within collective action and commons governance. Hardin framed the dynamic as a *tragedy of the commons* (Hardin, 1968), while Ostrom documented institutional conditions under which communities avert that tragedy (Ostrom, 1990). From a cognition viewpoint, our agent-level design follows the bounded-rationality tradition: people use fast, implementable heuristics adapted to their environments (Gigerenzer & Gaissmaier, 2011). At the system-level, designing those heuristics transforms an ill-structured problem (Simon, 1973) into well-structured subproblems with explicit objectives and evaluators.

**Mechanisms, nudges, and AI-personalized messaging:** Adoption is often the bottleneck, and even good policies underperform without mechanisms for uptake. Behavioral nudges and choice architecture can shift real-world environmental decisions (Byerly et al., 2018). Recent evidence shows that generative models can craft personalized messages with stronger persuasive effects than generic appeals (Matz et al., 2024; Rogiers et al., 2024). Our MIMIC phase operationalizes this by evolving messages that reliably alter agents' code-level heuristics toward ECHO-derived targets.

## 3 PROBLEM FORMULATION AND APPROACH

The collective action setting is ill-structured at two coupled levels. At the *agent level*, each agent $i \in \mathcal{N}$ observes a local state $S_{L,i}$ (e.g., their own resources, constraints, and immediate context), chooses an action $a_i \in \mathcal{A}_i$ (e.g., how much to extract, when to act, or where to intervene), and optimizes a local objective $U_{L,i}$ (e.g., profit, convenience, or personal cost). However, the effect of $a_i$ on societal goals depends on the unknown and evolving actions of others, $a_{-i}$. At the *system level* (policymaker), inferring what agents currently do (baseline behavior), determining how to coordinate local choices so they aggregate into desired global patterns, and how to incentivize behavior under real-world constraints are themselves ill-structured problems. We consider two domains: agricultural landscape management and carbon-aware EV charging coordination.

Let $\mathbf{A} = (a_1, \ldots, a_N)$ denote the joint action profile and define a nonseparable global objective $U_G(\mathbf{A}) = G(\Phi(\mathbf{A}))$, where $\Phi$ maps joint interventions to a mesoscale representation (e.g., a habitat graph or a load profile), and $G$ scores that representation. From any single agent's vantage point, $\partial U_G / \partial a_i$ depends on unknown and evolving $a_{-i}$ and is mediated by thresholds, complementarities, and path dependence in $\Phi$. These properties render one-shot mechanism design ill-posed. Therefore, to make this collective action ISP tractable, our approach imposes structure at both the system and agent levels by decomposing the problem into four well-structured stages whose outputs are executable (Fig. 1):

**Stage 1 - Establish Baseline Behavior:** We fix what agents do by default. For each agent $i$, we compute the baseline action $a_i^0$ by solving the local problem $\max_a U_{L,i}(a, S_{L,i})$. This yields state-action pairs $D_i^0 = (S_{L,i}, a_i^0)$. Practically, this means solving for (or recording) each agent's personal choices, for example, how much to ecologically intervene on the agricultural farm or how much to use a particular time slot for charging through the week.

**Stage 2 - Learn Baseline Heuristics:** We learn executable code heuristics $\hat{H}_{L,i}$ that reproduces $a_i^0$ from $S_{L,i}$, where candidates are Python programs. An LLM proposes/mutates code and an EA selects by a computable error between predicted and baseline actions. In effect, this yields a program for each agent that, given the observables, outputs the same local actions the agent would normally choose using personal preferences.

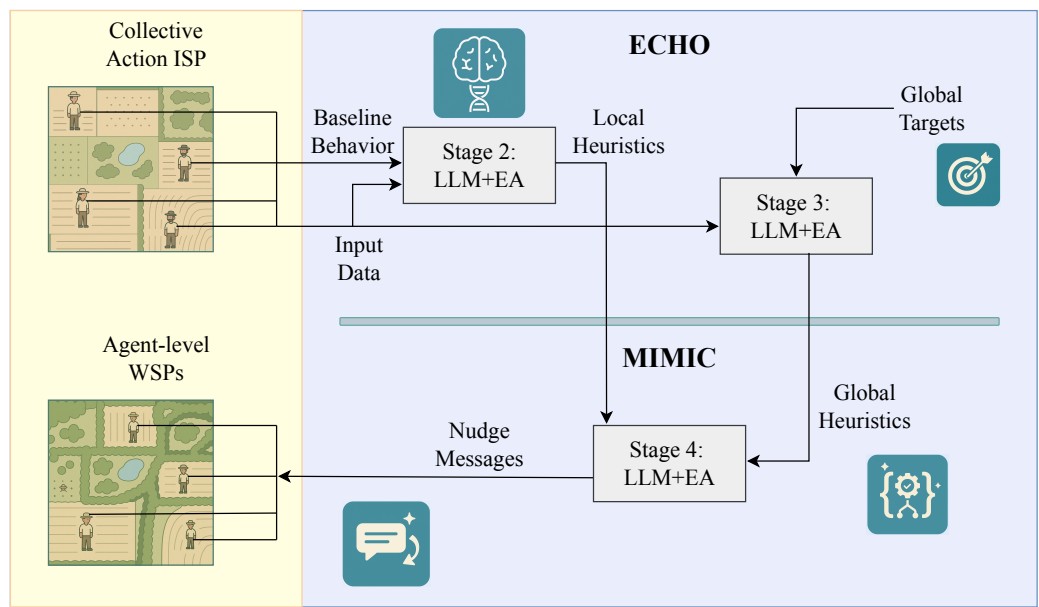

Figure 1: **ECHO–MIMIC framework**. ECHO uses an LLM+EA search loop to propose personal-level decision heuristics aligned with baseline (stage 2) and global (stage 3) objectives. MIMIC optimizes personalized nudges (e.g., messages/mechanisms/policies) using an LLM+EA search loop to drive collective action. Overall, the framework converts the collective action ISP into system- and agent-level WSPs. Figure uses the farm domain. See Appendix B.2 for a more detailed workflow.

**Stage 3 - Learn Global Heuristics:** We identify globally desirable targets (directions) $H^*_{G,i}$ that maximize $U_G$, then learn executable code $\hat{H}^*_{G,i}$ that maps $S_{L,i} \mapsto H^*_{G,i}(S_{L,i})$. Candidates are again Python heuristics evolved by LLM+EA and scored by an appropriate fitness score. In our domains, this produces programs that collectively improve landscape connectivity or clean-energy charging.

**Stage 4 - Nudge to Global Heuristics:** We discover natural-language messages $M_i$ that move agents from executing $\hat{H}_{L,i}$ toward $\hat{H}^*_{G,i}$. In simulation, an Agent LLM seeded with the code of $\hat{H}_{L,i}$ and a persona, receives a message from a Policy LLM, edits its code to a temporary $H_{\text{nudged},i}$ if persuaded, and outputs an action $\tilde{a}_i$. Fitness rewards messages that make $\tilde{a}_i$ close to $\hat{H}^*_{G,i}$.

For the policymaker, these four stages are WSPs, with finite candidate sets and computable fitness. For agents, scripts $\hat{H}_{L,i}$ and $\hat{H}^*_{G,i}$ are executable, and messages $M^*_i$ minimize cognitive burden.

## 4 THE ECHO–MIMIC FRAMEWORK

ECHO–MIMIC is an end-to-end framework to think about collective action problems. We first start by breaking down any collective action problem into the four stages discussed in the previous section. We then implement the four stage decomposition in two coupled phases. First, ECHO discovers executable heuristics (Stages 2–3), followed by MIMIC, which discovers mechanisms to adopt them (Stage 4). Both phases follow the same design philosophy: the LLM serves as the variation engine, generating, mutating, crossing over, repairing, and reflecting on candidates, while the Evolutionary Algorithm supplies selection pressure via computable fitness.

### 4.1 ECHO: EVOLUTIONARY CRAFTING OF HEURISTICS FROM OUTCOMES

ECHO learns executable Python heuristics that replicate baseline local behavior ($\hat{H}_{L,i}$, Stage 2) and globally desirable behavior ($\hat{H}^*_{G,i}$, Stage 3). Each candidate is a constrained function with a fixed I/O signature that reads $S_{L,i}$ and returns actions.

To implement this phase, we evolve a population of $K$ candidates for $H$ generations using three LLM roles. These include a *Generator* to produce initial population of programs, a *Modifier* to apply mutation, crossover, and reflect-and-improve edits, and a *Fixer* to repair compile/runtime issues in programs. The Modifier and Fixer are used in tandem each round, followed by elitism to preserve the top-$k$ candidates. Stage 2 and 3 use distinct fitness functions. In stage 2, the fitness minimizes the error between a candidate's action and the baseline $a_i^0$, yielding $\hat{H}_{L,i}$ as explicit approximations to locally rational behavior. Whereas in stage 3, the fitness minimizes the error between a candidate's action and the global targets $H_{G,i}^*$, returning $\hat{H}_{G,i}^*$ as policies for the collective goal.

### 4.1.1 Prompting Design and Neighbor In-Context Learning in ECHO

**Generator LLM:** To propose an initial population of executable heuristics with the required I/O signature, we compose the prompt as

$$\mathcal{P}^{\text{gen}} = [\text{System}] \oplus [\text{Task}] \oplus [\text{ICL}_{\mathcal{N}(i)}] \oplus [S_{L,i}] \oplus [\Theta],$$

where $\oplus$ refers to concatenation; [System] fixes coding constraints and file I/O; [Task] restates the goal of returning proper actions and failure modes to avoid; [ICL$_{\mathcal{N}(i)}$] is a small set of (input, output) exemplars from neighbors $\mathcal{N}(i)$ for in-context learning (ICL); [$S_{L,i}$] is the current agent's feature vector. [$\Theta$] collects other global parameters (prices, costs etc.).

**Choosing neighbors for ICL:** We define $\mathcal{N}(i)$ as $k$ adjacent farms, and supply examples

$$\left\{ \left( \text{GeoJSON}_j^{\text{in}}, \text{GeoJSON}_j^{\text{out}} \right) \right\}_{j \in \mathcal{N}(i)}$$

summarizing state and the realized interventions. This introduces the model to patterns likely to transfer under similar geographical and social conditions. Neighbor ICL allows us to withhold the current agent's baseline labels to test whether the LLM can infer decision rules from analogous contexts when supervision is provided indirectly via EA selection. It also mirrors observational diffusion in society, where practices propagate through local networks facing shared pressures.

**Modifier LLM:** For genetic variation operators in the evolutionary loop, we use

$$\mathcal{P}^{\text{mod}} = [\text{System}] \oplus [\text{Task}] \oplus [\text{Operator}] \oplus [\Theta] \oplus [\text{Candidates}],$$

where [Operator] specifies the details regarding the operation to be performed (mutate, crossover, reflect, see Appendix B.3), and [Candidates] includes the parent(s) and, for *reflect*, a brief leaderboard with fitness scores. [System] and [Task] are similar to the ones used for generation.

**Fixer LLM:** When a candidate triggers compile/runtime errors, the Fixer LLM performs minimal edits to restore validity while preserving the required I/O signature and intended behavior.

### 4.2 MIMIC: Mechanism Inference & Messaging for Individual-to-Collective Alignment

MIMIC is designed to imitate a central planner that coordinates between agents by observing their locally optimal heuristics, computing their potentially globally optimal heuristics, and using this information to change their behavior in the right direction. To do so, it searches for natural-language mechanisms $M_i$ that reliably steer agents from running $\hat{H}_{L,i}$ toward $\hat{H}_{G,i}^*$ (Stage 4). The population is textual candidates made of economic incentives, behavioral framings, and hybrids, generated/-modified by *Policy LLMs*. Each message is evaluated in a simulation with an *Agent LLM* (Farmer, EV Owner) that is initialized with both a persona and the program $\hat{H}_{L,i}$. To ensure robust evaluation, we construct agent personas by using traits relevant to the domain, which drive the agent's decision process. We also implement an explicit refusal mechanism, where if a proposed message conflicts with the agent's core values or constraints (as defined by its persona), the agent can reject the message and stick to its baseline heuristic $\hat{H}_{L,i}$. Therefore, upon reading $M_i$, the Agent LLM may propose edits to its code or make no changes, yielding $H_{\text{nudged},i}$, which outputs an action $\tilde{a}_i$. Fitness rewards messages that make $\tilde{a}_i$ closely match $\hat{H}_{G,i}^*$ (Appendix B.2; Fig. 5b). Thus, MIMIC is effective because its objective is defined against ECHO's executable heuristics and persuasion is measured as concrete code edits that change behavior.

We us the following LLMs in MIMIC to perform different actions:

**Policy Generator LLM:**  To propose candidate nudges, the Policy Generator composes

$$\mathcal{P}^{\text{pol-gen}} = [\textsc{System/Framing}] \oplus [\textsc{Task}] \oplus [\textsc{DecisionContext} : S_{L,i}, \hat{H}_{L,i}, \hat{H}_{G,i}^*, \Theta]$$
$$\oplus [\Theta^{\text{mech}}],$$

where $[\Theta^{\text{mech}}]$ encodes mechanism constraints (e.g., budget caps). The model outputs a structured $M_i$ tailored to the persona with framing as instructed.

**Policy Modifier LLM:**  Given parent messages, the Policy Modifier applies constrained edits via

$$\mathcal{P}^{\text{pol-mod}} = [\textsc{System}] \oplus [\textsc{Operator}] \oplus [\textsc{DecisionContext}] \oplus [\Theta^{\text{mech}}] \oplus [\textsc{Candidates}],$$

and returns $M_i'$ that preserves constraints (budget honesty, no coercive framing) while increasing persuasion, measured downstream by induced $(H_{\text{nudged},i}, \tilde{a}_i)$ and fitness against $\hat{H}_{G,i}^*(S_{L,i})$.

**Agent (Simulation) LLM:**  We emulate an agent's response to candidate nudges with an Agent LLM. The prompt is composed as

$$\mathcal{P}^{\text{sim}} = [\textsc{System/Persona}] \oplus [\textsc{DecisionContext} : S_{L,i}, \hat{H}_{L,i}, \Theta] \oplus [\textsc{Message} : M_i],$$

where $[\textsc{System/Persona}]$ fixes background, goals, and receptivity; $[\textsc{DecisionContext}]$ specifies the local state $S_{L,i}$, the baseline heuristic $\hat{H}_{L,i}$, and constraints/parameters; and $[\textsc{Message}]$ is the candidate nudge from the Policy LLMs. The model returns $H_{\text{nudged},i}$ which when executed gives $\tilde{a}_i$, tying persuasion to code edits and actions that can be scored against $\hat{H}_{G,i}^*$.

To summarize, we use ECHO to discover *what* to do and MIMIC to discover *how* to get people to do it. This coupling turns a challenging ISP into a chain of WSPs whose outputs are deployable, i.e., communicate $M_i^*$ to each agent to induce adoption of $\hat{H}_{G,i}^*$. Full prompt templates for the stages, LLM roles, operators, and personas for the farm domain are given in Appendix F.

### 4.3 Automated Domain Creation Agent

To enable our framework to generalize across domains without manual prompt engineering, we introduce a Domain Creation Agent. This agent takes as input a high-level domain schema of: a) *Agent State ($S_{L,i}$)*: Description of local variables (e.g., crop yields, charging demand). b) *Action Space ($a_i$)*: Allowable decisions (e.g., intervention length, slot usage). c) *Observability*: What neighbors or global signals are visible. d) *Constraints*: Budget, physical limits, or regulatory bounds. Using a meta-prompt, the agent generates the specific system instructions, task prompts, and evaluation harness for the ECHO and MIMIC stages. This ensures that the prompt and evaluation templates are modular and composable, automatically adapting to the specific terminology and logic of the new domain, allowing our framework to scale to new collective action problems.

## 5 Experimental Results

We demonstrate the application of our ECHO-MIMIC framework on two distinct collective action domains: agricultural landscape management and carbon-aware EV charging coordination.

**Agricultural Landscape Management:** In this domain, we follow models of ecological intensification (Kremen, 2020; Bommarco et al., 2013; Dsouza et al., 2025) where biodiversity outcomes hinge on spatial configuration (Taylor et al., 1993). Each agent $i$ (farmer) observes local state $S_{L,i}$ consisting of plot-level agro-ecological and economic features (crop types, yields, prices). Actions $a_i$ are farm interventions: (i) *margin intervention* (length, placement), and (ii) *habitat conversion* (area, orientation). The local objective $U_{L,i}$ is net present value (NPV) under farm-specific constraints, while the global objective $U_G$ prioritizes landscape-scale ecological connectivity, measured by the *Integral Index of Connectivity (IIC)* (Pascual-Hortal & Saura, 2006). We simulate an agricultural landscape of 5 farms (Fig. 2a) by generating synthetic farm and plot-level geo-spatial data based on real farm data from the 2022 Canadian Annual Crop Inventory (CACI) (Agriculture and Agri-Food Canada (AAFC), 2022).

**Carbon-Aware EV Charging Coordination:** In this domain, which models the challenge of coordinating distributed energy resources (Anderson et al., 2023; Cheng et al., 2022), each agent $i$ (EV

Table 1: Mean accuracy (averaged over 5 agents and 2 seeds per domain) for ECHO–MIMIC, DSPy MIPROv2, and AutoGen across two domains (Farm, EV) and five models. ECHO (Stage 2+3) and MIMIC (Stage 4) together form the ECHO-MIMIC pipeline. The evolutionary algorithm is configured with a population of 25 individuals and run for 25 generations. G2.0-FT is omitted for the EV domain and AutoGen due to lack of reliable API access. Models: G2.0-FT = Gemini 2.0 Flash Thinking, G2.5-F = Gemini 2.5 Flash, G2.5-P = Gemini 2.5 Pro, GPT5-n = GPT-5 nano (medium), GPT5-m = GPT-5 mini (medium).

| Domain | Stage | Method | G2.0-FT | G2.5-F | G2.5-P | GPT5-n | GPT5-m |
|--------|-------|--------|---------|--------|--------|--------|--------|
| | 2 | DSPy MIPROv2 | 0.41 | 0.46 | 0.53 | 0.45 | 0.55 |
| | 2 | ECHO | 0.93 | 0.94 | 0.95 | 0.94 | 0.95 |
| | 2 | AutoGen | – | 0.40 | 0.47 | 0.43 | 0.52 |
| | 3 | DSPy MIPROv2 | 0.00 | 0.00 | 0.12 | 0.00 | 0.18 |
| Farm | 3 | ECHO | 0.24 | 0.29 | 0.33 | 0.27 | 0.35 |
| | 3 | AutoGen | – | 0.08 | 0.10 | 0.05 | 0.14 |
| | 4 | DSPy MIPROv2 | 0.33 | 0.35 | 0.43 | 0.38 | 0.43 |
| | 4 | MIMIC | 0.73 | 0.75 | 0.79 | 0.71 | 0.82 |
| | 4 | AutoGen | – | 0.33 | 0.44 | 0.37 | 0.46 |
| | 2 | DSPy MIPROv2 | – | 0.51 | 0.60 | 0.58 | 0.62 |
| | 2 | ECHO | – | 0.95 | 0.96 | 0.95 | 0.97 |
| | 2 | AutoGen | – | 0.39 | 0.50 | 0.44 | 0.48 |
| | 3 | DSPy MIPROv2 | – | 0.66 | 0.68 | 0.67 | 0.71 |
| EV | 3 | ECHO | – | 0.87 | 0.91 | 0.85 | 0.93 |
| | 3 | AutoGen | – | 0.38 | 0.47 | 0.40 | 0.44 |
| | 4 | DSPy MIPROv2 | – | 0.70 | 0.75 | 0.72 | 0.76 |
| | 4 | MIMIC | – | 0.91 | 0.93 | 0.91 | 0.94 |
| | 4 | AutoGen | – | 0.78 | 0.82 | 0.79 | 0.83 |

owner) observes local state $S_{L,i}$ consisting of base demand across time slots, preferred charging slots, and comfort penalties for non-preferred slots. Actions $a_i$ are daily usage vectors (one per slot). The local objective $U_{L,i}$ minimizes electricity price and comfort penalties, while the global objective $U_G$ minimizes carbon emissions, grid overload, and slot-usage constraints. We generate synthetic scenarios with 5 agents, 4 time slots, and 7-day horizons, with varying input data. See Appendix B.7 for more info on data generation for both domains.

## 5.1 ECHO-MIMIC Outperforms Baselines at Driving Collective Action

As there is no direct comparison to ECHO-MIMIC driving collective action by working at both the system and agent levels, we assume system level breakdown into stages, and compare at the agent level against DSPy MIPROv2 (Opsahl-Ong et al., 2024), a strong LLM-native baseline, and Auto-Gen (Wu et al., 2024), a general multi-agent framework (Table 1). We do not compare to non-LLM program search as our goal is not merely to approximate a global planner but to induce human-readable heuristics that can be executed by agents and seamlessly verbalized into messages. Across both domains, ECHO-MIMIC outperforms both DSPy and AutoGen in all stages under identical input constraints. DSPy struggles to induce global-compatible local heuristics in stage 3, while AutoGen, lacking the explicit evolutionary pressure on code/message structure, fails to consistently discover high-performing policies. These results show consistent cross-domain and cross-LLM gains of ECHO-MIMIC in generating executable heuristics and messages, beyond what generic LLM program-synthesis or agent frameworks achieve. Finally, we noticed that though baselines perform better with more capable LLMs (G2.5-P, GPT5-m), their performance cannot match ECHO-MIMIC, which also benefits from higher capability.

## 5.2 ECHO Discovers Context-Aware Heuristics

ECHO reliably evolves Python heuristics that approximate local behavior across heterogeneous agents. In the farm domain, ECHO learns when to choose margin versus habitat conversion at

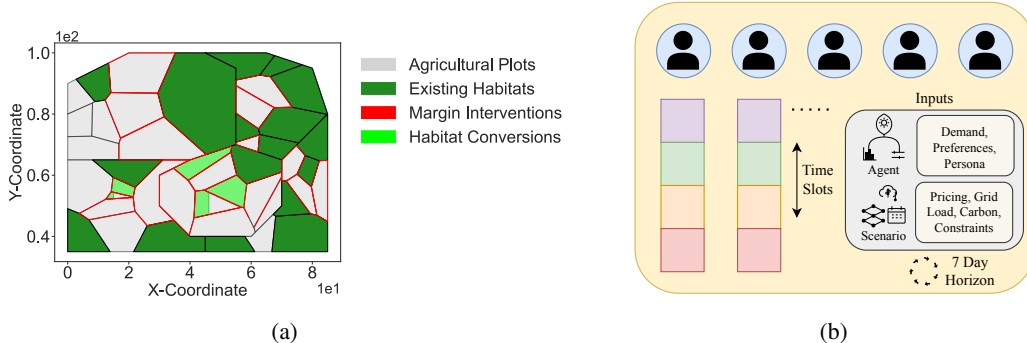

(a)                                                        (b)

Figure 2: **Domain specific actions**. a) Interventions resulting from ECHO learned baseline heuristics in stage 2 for the farm domain. The interventions match the ground-truth baseline computed from stage 1 closely. For the comparison with ground-truth, ECHO stage 3 predictions, and synthetically generated farm geometries, see Appendix C. b) Synthetically generated EV charging spatiotemporal configuration. Five agents are placed in a line, each with their own charging demand, preferences, and carbon intensity. They are allowed to specify usage in four time slots for a week.

the plot level (Fig. 2b), improving fitness across generations for all farms (Fig. 4a). Farms 2 and 5 converge quickly, while Farms 1, 3, and 4 improve more gradually, indicating harder optimization landscapes. Lineage analysis of the best final heuristics for both the farm and EV charging domains show *Crossover* is both the most frequent operator and the largest contributor to cumulative fitness gains (Fig. 4b). *Mutate* is also common and adds steady improvements. *Reflect* appears infrequently in top lineages and adds little directly, suggesting it supports diversity rather than breakthroughs (Appendix C; Fig. 11b).

Across agents, fitness typically rises with code-complexity indicators (e.g., logical lines of code, Halstead difficulty, distinct (H1) operators up to an intermediate optimum; beyond that point, additional complexity correlates with lower fitness. We plot this phenomenon for the farm domain in Fig. 9c (also see Appendix C; Fig. 10). Maintainability tends to decline as fitness rises, consistent with more intricate logic being leveraged to capture hard cases. Farm 3, 4 show particularly steep gains at higher distinct-operator counts, suggesting that richer program vocabularies are necessary to escape performance plateaus (Fig. 9c, Appendix C; Fig. 10). On Farm 3, adding prompt instructions that explicitly encourage high Halstead distinct-operator counts and difficulty produces consistently higher accuracy, with a clear divergence after generation 15 (Fig. 9d). This indicates that seeding the search with more expressive building blocks expands the recombination space that operators can exploit later in evolution.

Evolved programs implement multi-layered logic, for instance in the EV charging domain, by calculating headroom safety and determining the allocation based on persona like below:

```
Inputs: capacity, baseline, base_demand, carbon, tariff
Outputs: usage_allocation, preference_score

headroom <- capacity - baseline - base_demand
preference_score <- carbon + (tariff * 1000)

if rationed_day is True and slot == 2:
    headroom <- headroom - 2.0
if headroom > 0.05:
    allocatable <- headroom - 0.05
if remaining_load > 0:
    usage_allocation <- min(remaining_load, allocatable)
    remaining_load <- remaining_load - usage_allocation
```

Other representative heuristics can be found in Appendix D, highlighting ECHO's ability to integrate economic and spatial reasoning like computing tariff-weighted exponential demand or polygon orientation via PCA. In summary, ECHO discovers context-aware heuristics in both domains, operators play distinct roles, and controlled increases in code complexity can unlock superior performance.

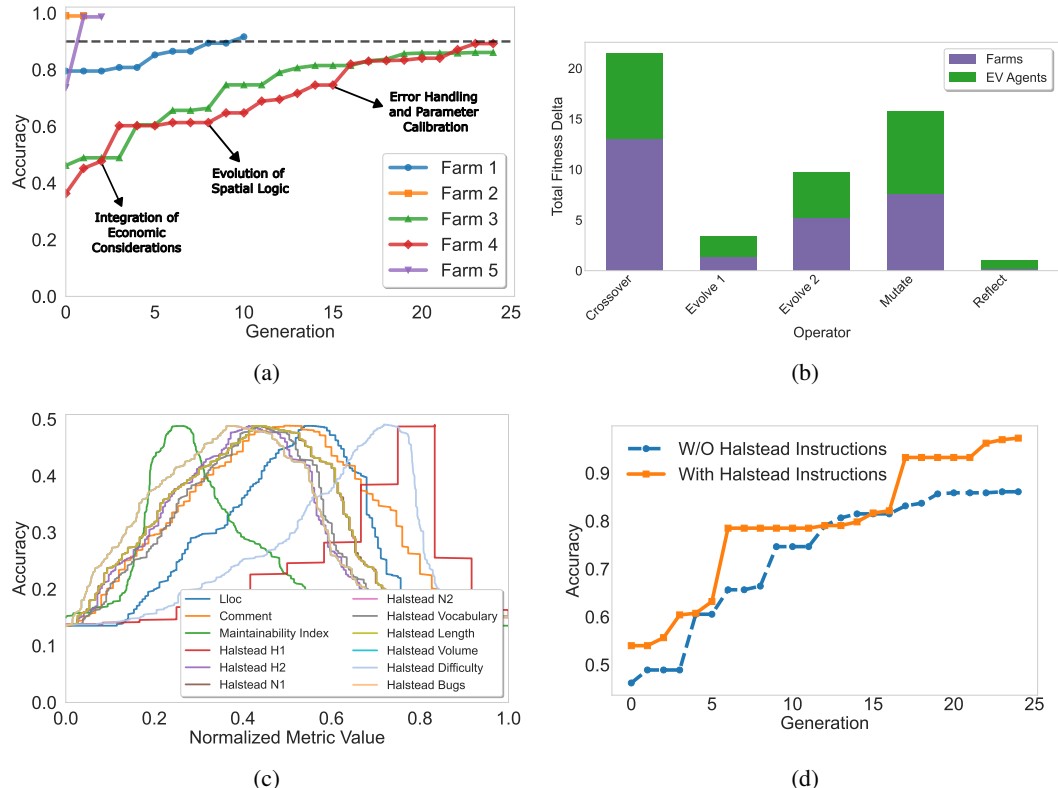

Figure 3: **ECHO stage 2 results**. a) Accuracy $(1 - error)$ over generations in stage 2 for the farm domain. Some farms are easier to make progress in (farms 2, 5) compared to others (farms 1, 3, 4), and distinct capabilities emerge in harder farms as generations progress. b) Total fitness $(1/error)$ delta for both the domains together, resulting from LLM variation operators, summed across generations for the best performing program at the end. *Crossover* and *mutate* have the highest positive cumulative change in fitness. c) Accuracy versus normalized complexity metrics of the heuristics for farm 3 in the farm domain. Increased Halstead metrics are correlated with increased accuracy, upto a point, followed by a decrease. d) Accuracy over generations with and without Halstead instructions for farm 3 in the farm domain. Adding additional Halstead instructions to the prompt provides free gains in accuracy at the expense of interpretability.

### 5.3 MIMIC EVOLVES PERSONALITY-ALIGNED NUDGES

LLMs can produce persuasive text that draws on behavioral science to scale tailored messages (Matz et al., 2024; Rogiers et al., 2024). Yet nudge efficacy is highly context-dependent and hard to evaluate. MIMIC addresses this with a closed-loop search between two agents: a *Policy LLM* that generates candidate nudges and an *Agent LLM* that simulates agent responses and executes heuristics.

In the EV charging domain, with versatile personas (to model agent heterogeneity) and generic instructions (no specific framing), accuracy with respect to generated global heuristic actions from ECHO (stage 3) improves across generations and agents (Fig. 4a). In the farm domain we use three personas, *Resistant*, *Economic*, and *Social*, and two nudge types, *Economic* and *Behavioral* (choice-architecture levers such as social comparison, defaults, commitments, and framing (Byerly et al., 2018; Carlsson et al., 2021)). We see that social personas + behavioral nudges, and economic personas + economic nudges, perform the best (Fig. 4b), while economic personas also benefit from behavioral nudges after an initial lag. Both these experiments demonstrate the persona and framing specific targeting potential of MIMIC. Qualitatively, across both domains, we see that top behavioral nudges leverage social proof and low-risk trials, while top economic nudges offer subsidies/premiums with clear commitments. Full best-message exemplars are in Appendix G. In summary, MIMIC

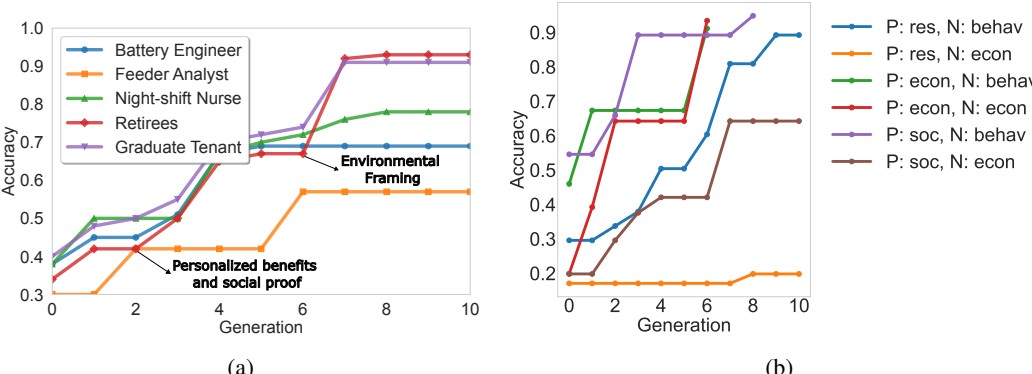

(a)  (b)

Figure 4: **MIMIC nudge discovery and personalization**. Accuracy of nudges over generations a) for the EV charging domain with versatile personas and generic instructions. At different points MIMIC learns to use personalized benefits, social proof, and environmental impact framing. b) for the farm domain (hard farms) with persona and nudge type specific instructions. P refers to personas and N refers to nudge types. Personas are either Resistant, Economic, or Social. Nudge Types are either Behavioral or Economic.

adapts collective action nudges to context and persona, while traditional static mechanisms and purely economic incentives struggle with such heterogeneity (Knowler, 2014). Moreover, MIMIC (together with ECHO) is readily extensible to human-in-the-loop deployment, where real feedback replaces simulated responses for iterative refinement (Appendix E).

## 6  DISCUSSION AND FUTURE WORK

We introduced ECHO-MIMIC, a general end-to-end framework that addresses ill-structured collective action by converting the system-level design problem into a sequence of well-structured searches for the policymaker and by producing executable heuristics that render each agent's local decision a WSP. Across both our agriculture and EV charging domains, ECHO learns heuristics that reproduce both personal preferences and globally important objectives. MIMIC then discovers messages that induce agents to adopt those executable targets. Together, these phases evolve *what* should be done and *how* to get it done, suggesting a practical path to scalable, adaptive policy design. Finally, our domain creation agent, by taking in input-output schema, observability, constraints, and domain specific details and automatically adapting the logic of the any domain to our framework, allows extension of our framework to any arbitrary collective action problem.

Despite the potential applications, there are some limitations of our current framework. First, the agent simulation abstracts human behavior. Personas and code-edit responses by a Farm LLM are proxies that require validation with real participants. Second, non-stationarity of prices, ecology, and policy can quickly stale learned heuristics and nudges. Distribution shift undermines both ECHO's scripts and MIMIC's messages. Third, persuasive mechanisms risk manipulation, unequal burden sharing, or disparate impacts on smallholders. Respecting privacy, transparency, and consent from the outset are essential. Finally, evolution can produce complex heuristics with deep branching and opaque feature engineering that erode interpretability/trust and create implementation frictions. This can potentially be alleviated by regularizing code complexity and enforcing functional signatures. Given these limitations, we see several directions for future work (see Appendix E.5 for more):

**Field validation:** conduct preregistered behavioral experiments and pilots with farmers to estimate heterogeneous treatment effects of nudge messages and to measure sim-to-real gaps.

**Online iterative refinement with real-world feedback:** although the EA selects high-fitness messages in simulation for each persona, post-deployment we can treat each rollout as a new generation and update the message and heuristic pool using real outcomes. See Appendix E for more details.

**Interpretability of heuristics:** curb complexity creep by adding complexity regularizers (e.g., functional signatures, MDL-style penalties, cyclomatic-complexity caps) and enforcing edit budgets.

## 7 ETHICS STATEMENT

Our study uses only synthetically generated data and simulated agents; no human participants, personally identifiable information, or proprietary private data were collected or analyzed. The synthetic data were procedurally generated, as detailed in Appendix B.7. We evaluate policy *nudges* exclusively in simulation via predefined agent personas and a closed-loop interaction between a Policy LLM and an Agent LLM; we note that these are proxies and call for preregistered field studies before any deployment. To mitigate foreseeable risks (e.g., manipulation, unequal burdens, privacy harms, or distribution-shift failures), we propose governance measures, human-in-the-loop approvals, privacy-preserving telemetry and opt-in consent, as outlined in Appendix E.4. We also discuss value-laden choices and Goodhart risks of proxy objectives and recommend stress-testing and transparency (Appendix E.5). Any funding or affiliations will be disclosed in the paper's acknowledgments.

## 8 REPRODUCIBILITY STATEMENT

We provide an anonymous supplementary zip with all source code to reproduce results. The paper and appendix specify model choices (e.g., Gemini variants and evolutionary settings) and libraries/interfaces used, enabling replication of LLM-EA runs (Appendix B.1). Execution occurs in a controlled environment (json/numpy/shapely I/O from input.geojson to output.*) with comprehensive logging of fitness scores, operator usage, candidate trajectories, and code-complexity metrics, details that support exact reruns and diagnostics (Appendix B.4). Fitness definitions for all stages (local/global heuristics and nudging) are formalized in B.5 with explicit error metrics, and the fitness-evaluation loop is diagrammed (Figs. 5,6) for clarity. Data generation is fully specified in B.7, enabling others to rebuild the synthetic datasets. Finally, we include representative heuristic programs (Appendix D) and complete prompts (Appendix F) to aid verification.

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

# A SUBMISSION DETAILS

## A.1 SOURCE CODE

Source code associated with this project is attached as a supplementary zip file.

## A.2 Use of Large Language Models

We used large language models (LLMs) in the following scoped, human-supervised ways: (i) Writing polish. Draft sections were refined for clarity, structure, and tone; all technical claims, numbers, and citations were authored and verified by us, and every LLM-suggested edit was line-reviewed to avoid introducing errors or unsupported statements. (ii) Retrieval & discovery. We used LLMs to craft and refine search queries to find related work and background resources; candidate papers were then screened manually, with citations checked against the original sources to prevent hallucinations. (iii) Research ideation. We used brainstorming prompts to surface alternative baselines, ablation angles, and failure modes; only ideas that survived feasibility checks and pilot experiments were adopted. (iv) Coding assistance (via Cursor, Gemini, and OpenAI). We used Cursor's inline completions and chat for boilerplate generation (tests, docstrings, refactors); We used Gemini-2.5-pro and o3 to generate code snippets for different parts of the project; all code was reviewed before inclusion. Across all uses, we ensured that LLM outputs never replaced human analysis, reproducibility artifacts, or empirical validation.

# B Implementation Details

## B.1 Models

Our experimental setup leverages *gemini-2.0-flash-thinking-exp-01-21*, *gemini-2.5-flash*, *gemini-2.5-pro*, *gpt-5-nano*, and *gpt-5-mini* models for the core tasks of heuristic generation, modification, fixing, and agent simulation. We compare our method and baselines across these family of models. The evolutionary algorithm was configured with a population size of 25 individuals and was run for a maximum of 25 generations for ECHO and 10 generations for MIMIC.

## B.2 Overall ECHO-MIMIC Workflow and Components

Fig. 5 summarizes the complete ECHO-MIMIC pipeline. In ECHO (Fig. 5a), an LLM–evolutionary loop proposes, executes, and scores human-readable farm heuristics against baseline (Stage 2) and global (Stage 3) objectives under the same observability constraints used at deployment. In MIMIC (Fig. 5b), the system translates the learned heuristics into actionable nudges: messages/mechanisms/policies, then simulates agent responses to iteratively refine adoption.

The two reusable building blocks are detailed in Fig. 6: a robust fitness-evaluation-and-repair loop that executes candidate programs on farm data, scores outcomes, and attempts automatic fixes on failures (Fig. 6a), and an LLM-driven variation engine with mutation, crossover, exploration, and reflection operators to generate improved candidates across iterations (Fig. 6b). Together, these components enable end-to-end search over interpretable heuristics and their message-level implementations while preserving decision-time observability constraints.

## B.3 LLM-Guided Evolutionary Operators

The evolutionary search in both the ECHO and MIMIC phases is driven by a set of variation operators executed by a *Modifier LLM*. These operators take one or more parent candidates from the population and generate a new offspring candidate.

- **Mutation:** The LLM receives a single parent candidate (either a Python script or a natural language message) and is prompted to introduce a subtle mutation aimed at improving performance while preserving the core structure and validity of the candidate.

- **Crossover:** The LLM is given two parent candidates and prompted to combine them in an optimal way to cover heuristics/information from both. The goal is to produce a child that synergistically integrates advantageous traits from both parents.

- **Exploration 1 (Diverge):** Given two parents, the LLM is prompted to generate a new candidate that is as different as possible to explore new ideas. This operator encourages diversification and prevents premature convergence by exploring novel regions of the search space.

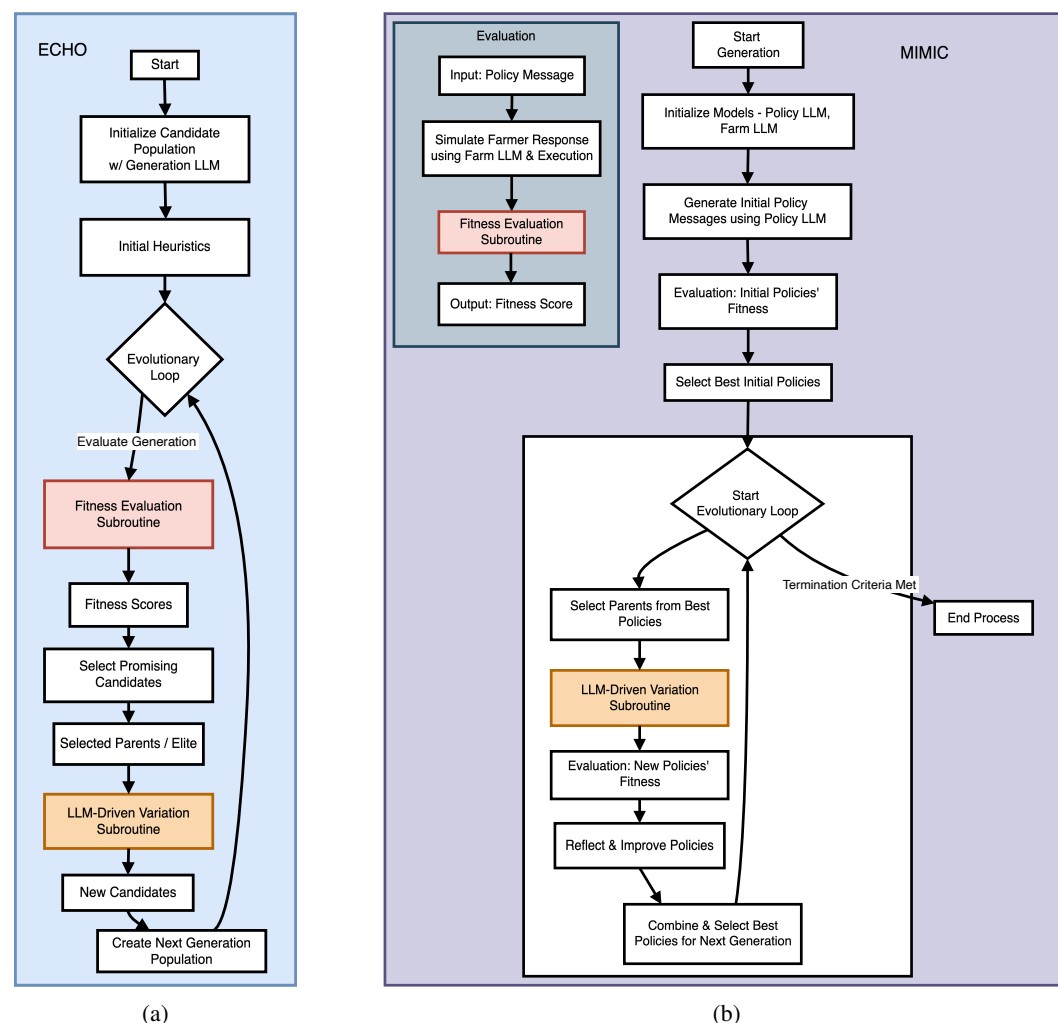

(a)                                   (b)

Figure 5: **ECHO–MIMIC framework**. a) ECHO uses an LLM-evolutionary search loop to propose, score, and select farm-level decision heuristics aligned with baseline (stage 2) and global (stage 3) objectives. (b) MIMIC optimizes personalized nudges (e.g., messages/mechanisms/policies) using an LLM-evolutionary search loop, evaluates nudges using simulated agent responses, and iteratively updates nudges to drive collective action. Illustration uses the farm domain as an example. See Fig. 6 showing the two subroutines of fitness evaluation and LLM-driven variation.

- **Exploration 2 (Converge & Innovate):** The LLM receives two parents, identifies common ideas between them, and then designs a new candidate based on these shared concepts but also introduces novel elements. This balances the exploitation of successful ideas with the exploration of new variations.

- **Reflection:** The LLM is provided with the top $k$ (e.g., 5) candidates from the current population, along with their fitness scores. It is prompted to analyze these heuristics/messages and craft a new one that is expected to have increased fitness. This allows the system to consolidate progress and make more informed, innovative leaps.

## B.4 ENVIRONMENT MANAGEMENT DETAILS

Some management, execution, and tracking details are given below:

**Selection:** After generating offspring through the evolutionary operators, a selection strategy determines which individuals proceed to the next generation. This involves methods like elitism (pre-

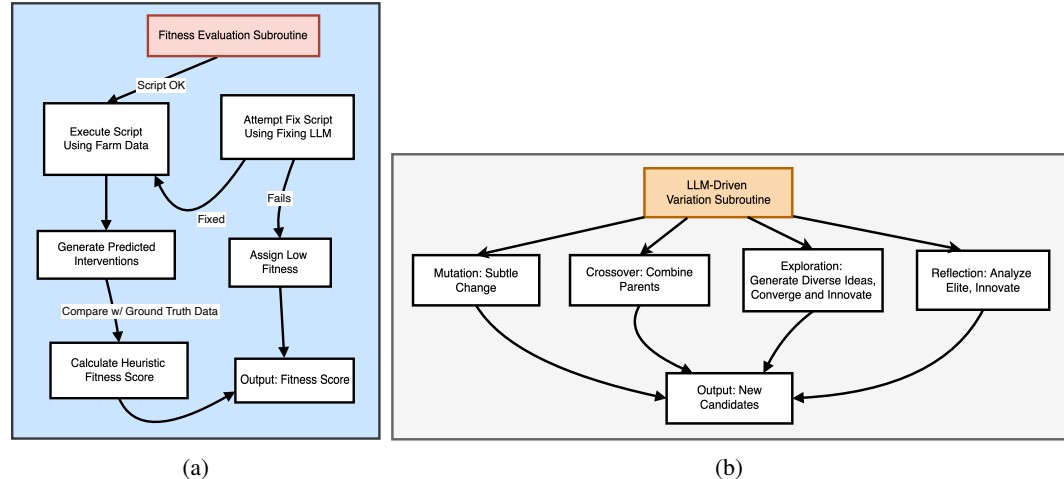

(a)  (b)

Figure 6: **LLM–EA candidate generation and fitness evaluation**. (a) Fitness evaluation & repair loop: run candidate scripts on data, generate predicted interventions, compare with ground-truth labels, and compute a heuristic fitness score; if execution fails, a Fixing-LLM attempts repair and unrepaired scripts receive low fitness, otherwise the repaired script is re-executed and scored, and the final fitness is output. Illustration uses the farm domain as an example. (b) LLM-driven variation subroutine: four operator families, mutation (subtle edits), crossover (combine parents), exploration (diverse ideas + converge & innovate), and reflection (analyze elites, innovate), produce new candidate heuristics/messages.

serving the best-performing individuals) combined with score-based selection from the combined pool of parents and offspring, maintaining a constant population size.

**Execution Environment:** Candidate Python scripts are executed in a controlled environment. This environment is equipped with necessary libraries such as json (for handling data files), numpy (for numerical operations), and shapely (for geometric operations). The scripts perform file I/O, reading from input.geojson and writing to output.geojson or output.json.

**Tracking:** Comprehensive data is logged for analysis and monitoring of the evolutionary process. This includes: fitness scores of all candidates, the representation of each candidate (Python code or natural language message), counts of how often each evolutionary operator is used, cumulative fitness deltas achieved by each operator, indicating their effectiveness, candidate trajectories, showing the sequence of operators applied to generate them, code complexity metrics (e.g., cyclomatic complexity, Halstead metrics) for Python script candidates, computed using the radon library. This helps in understanding the nature of the evolved solutions.

**Heuristics Explanation:** The generation of heuristic explanations follows a systematic, multi-stage pipeline (Fig. 7). The process involves an iterative loop which processes each Farm ID sequentially. For every farm, the core heuristic analysis begins by identifying and loading the relevant heuristic files. Concurrently, two LLMs are initialized, an Explanation LLM for generating initial explanatory summaries from code or data segments, and a Merge LLM for consolidating these explanations. The loaded heuristic files are subsequently processed in designated groups, typically consisting of three files each. As the system iterates through these file groups, the Explanation LLM analyzes the content of each group to generate an initial heuristic explanation. This newly generated explanation is then integrated into a cumulative summary. The Merge Model is then employed to combine the new group-specific explanation with the existing summary compiled from previous groups. Following this integration, the overall summary is updated, and an intermediate group summary is saved, allowing for checkpointing. Once all files for a given farm have been analyzed and their explanations merged, a final consolidated summary, representing the comprehensive heuristic explanation for that farm, is saved. The entire procedure concludes after this iterative processing has been completed for all designated Farm IDs. See "Heuristics Explanation" section in supplementary for more full prompts used for the two LLMs.

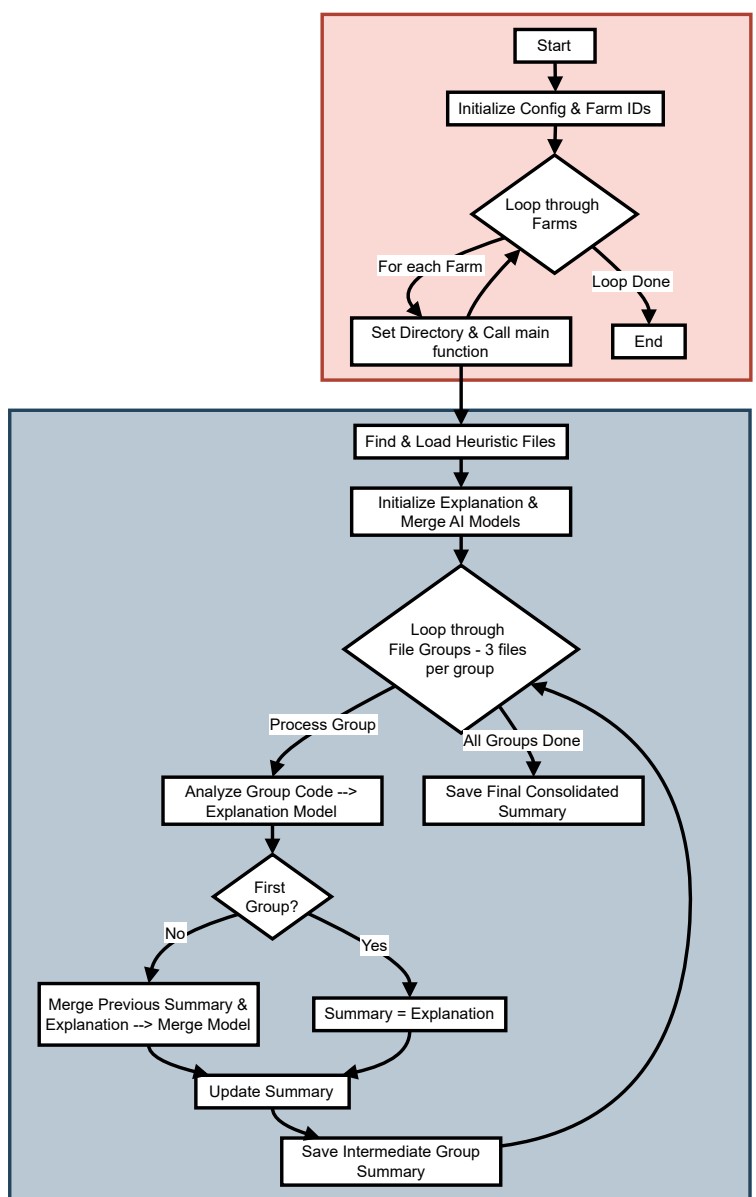

Figure 7: **Heuristic-explanation consolidation pipeline**. For each agent, initialize configs and load heuristic code files; then iterate over 3-file groups: an Explanation model analyzes each group to produce a draft, and, starting from the first group, a Merge model incrementally combines the running summary with each new explanation; intermediate group summaries are saved, followed by a final consolidated summary per agent. Illustration uses the farm domain as an example.

## B.5 FITNESS FUNCTION DETAILS

### B.5.1 AGRICULTURAL DOMAIN

Fitness for all candidates is calculated as the inverse of an error metric, with a small constant $\epsilon$ added to prevent division by zero: $Fitness = 1/(Error + \epsilon)$. Ground truth is obtained by computing results from existing ecological intensification and connectivity models (Dsouza et al., 2025).

**ECHO Fitness (Local Heuristics):** $Fitness_{NPV}$    The error is the Mean Absolute Error (MAE) between the intervention levels predicted by a candidate heuristic $(m_{p_i}, h_{p_i})$ and the ground-truth NPV-optimal levels $(m_{gt_i}, h_{gt_i})$ across all $N$ plots in a farm.

$$Error_{NPV} = \frac{1}{N} \sum_{i=1}^{N} (|m_{gt_i} - m_{p_i}| + |h_{gt_i} - h_{p_i}|)$$

**ECHO Fitness (Global Heuristics):** $Fitness_{CONN}$    The error is based on the Jaccard Distance between the sets of intervention directions predicted by the candidate $(MD_{p_i}, HD_{p_i})$ and the ground-truth connectivity-optimal directions $(MD_{gt_i}, HD_{gt_i})$.

$$JaccardDist(A, B) = 1 - \frac{|A \cap B|}{|A \cup B|}$$

$$Error_{CONN} = \frac{1}{N} \sum_{i=1}^{N} (JaccardDist(MD_{gt_i}, MD_{p_i}) + JaccardDist(HD_{gt_i}, HD_{p_i}))$$

**MIMIC Fitness (Nudging):** $Fitness_{NUDGE}$    The error measures the MAE between the intervention amounts produced by the agent's nudged heuristic $(m_{p_i}, h_{p_i})$ and the target fractional amounts derived from the global connectivity-optimal directions.

$$Error_{NUDGE} = \frac{1}{N} \sum_{i=1}^{N} \left( \left| \frac{|MD_{gt_i}|}{4} - m_{p_i} \right| + \left| \frac{|HD_{gt_i}|}{4} - h_{p_i} \right| \right)$$

### B.5.2 EV CHARGING DOMAIN

Fitness for all candidates is calculated as $1 - $ MAE (Mean Absolute Error) between the candidate's usage vector and the target usage vector, averaged across all days and slots.

$$Fitness = 1 - \frac{1}{D} \sum_{d=1}^{D} \frac{1}{S} \sum_{s=1}^{S} |u_{candidate}^{(d,s)} - u_{target}^{(d,s)}|$$

where $D$ is the number of days, $S$ is the number of slots per day, $u_{candidate}^{(d,s)}$ is the usage value of the candidate for day $d$ and slot $s$, and $u_{target}^{(d,s)}$ is the target usage value. The target usage vector varies by phase: for Local Heuristics, it is the local optimum; for Global Heuristics and Nudging, it is the global optimum.

## B.6 AGENT PERSONALITY AND NUDGE MECHANISM PROMPTS

In the MIMIC phase, the Farm LLM's persona and the Policy LLM's nudge generation are guided by specific system prompts.

- **Agent Personalities:** The system prompt for the Agent LLM establishes its background, goals, and receptiveness to advice. For example, the *Resistant* agent might be described as skeptical of new methods and valuing traditional practices, while the *Economic* agent is primarily focused on maximizing profit and return on investment. The *Social* agent is described as influenced by the actions of neighbors and community norms.
- **Nudge Mechanisms:** The Policy LLM is prompted to generate messages of a specific type. For an *Economic* nudge, the prompt might instruct it to design a financial incentive package within a budget that encourages adopting globally optimal practices. For a *Behavioral* nudge, the prompt instructs it to use principles like social proof, commitment, and framing to craft a persuasive message, without offering significant new economic incentives.

## B.7 DATA GENERATION

### B.7.1 AGRICULTURAL DOMAIN

To simulate agricultural landscapes, synthetic farm and plot-level geo-spatial data were generated (Fig. 8). The process began by establishing the combined boundaries of a farm cluster, which was then broken into five distinct farms using Voronoi tessellation based on random points; the resulting Voronoi cells were clipped to the edge to create a set of non-overlapping farms that together spanned the selected area. Each of these individual farm polygons was subsequently subdivided into nine land use plots, again using Voronoi tessellation, to produce smaller, non-overlapping plot polygons that filled the entire farm. Following this spatial design, properties were attached to each plot. First, plots were randomly divided into either agricultural plot (with a 60% probability) or habitat plot (40% probability). Later, a particular land use label was assigned through a weighted random strategy depending on its primary type: agricultural plots received crop type labels (e.g., Spring wheat, Oats, etc.) and habitat plots received land use type labels (e.g., Broadleaf, Grassland, etc.), with weights reflecting distributions from the 2022 Canadian Annual Crop Inventory (CACI) (Agriculture and Agri-Food Canada (AAFC), 2022). Following this, a yield value, drawn from a distribution matching the CACI data for the assigned crop, was matched to each agricultural plot. Ultimately, each synthetic farm's output was a GeoJSON FeatureCollection, detailing the geometric definitions (polygons) and the specific assigned attributes (type, label, yield) for every plot it contained.

### B.7.2 EV CHARGING DOMAIN

For the EV charging coordination domain, synthetic scenarios were generated with the following structure: 5 agents (EV owners), 4 time slots (representing different times of day), and 7-day planning horizons. Each agent was assigned a base demand profile (a 4-element vector representing charging needs across slots), a set of preferred charging slots (0-3 indices), a comfort penalty value (cost incurred when charging outside preferred slots), a persona, and a location on the grid feeder. Scenario-level parameters included slot-specific electricity pricing (varying by time of day), carbon intensity values (gCO2/kWh per slot), baseline grid load (non-EV load per slot), grid capacity limits, and slot-usage constraints (minimum and maximum number of agents allowed per slot). Multi-day profiles were created by varying these parameters across the 7-day horizon to simulate realistic temporal patterns (e.g., weekday vs. weekend pricing, weather-dependent carbon intensity). Neighbor in-context learning examples were constructed by sampling from other agents' configurations. All scenarios were serialized as JSON files containing agent configurations, daily profiles, and global parameters, enabling reproducible evaluation of evolved heuristics and nudges.

## B.8 DOMAIN CREATION AGENT

The Domain Creation Agent is a meta-level component designed to automate the adaptation of ECHO-MIMIC to new domains. It bridges the gap between a high-level problem description and the specific prompt templates required by the ECHO and MIMIC stages.

### B.8.1 WORKFLOW

1. **Input Schema**: The user provides a JSON-like schema defining the agent's state space, action space, and constraints.

2. **Meta-Prompting**: The Domain Creation Agent uses a meta-prompt that encodes the principles of good prompts (e.g., clear role definition and explicit constraints).

3. **Template Generation**: The agent generates:
   - *System Instructions:* Defines the role of the Policy LLM (e.g., "You are an expert in EV charging optimization...").
   - *Task Prompts:* Formats the specific state variables into a natural language description (e.g., "The battery is at 20%...").
   - *Operator Prompts:* Defines valid mutation operators for the code/text (e.g., "Change the threshold for urgent charging...").
   - *Evaluation Harness:* Generates scoring functions and JSON schemas based on the domain's objectives and evaluation criteria.

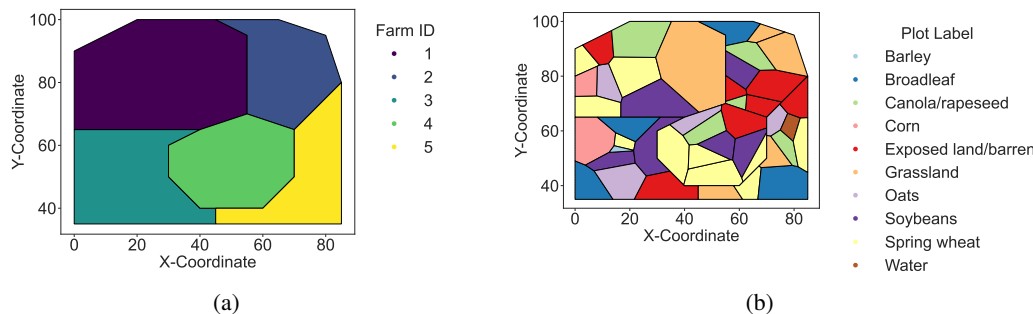

(a)                                             (b)

Figure 8: **Synthetic farms and plots**. a) Synthetically generated farm geometries and overall landscape configuration. Each farm is assigned its own distribution of crops, yields, and habitat plots. b) Each farm is assigned its own distribution of crops, yields, and habitat plots.

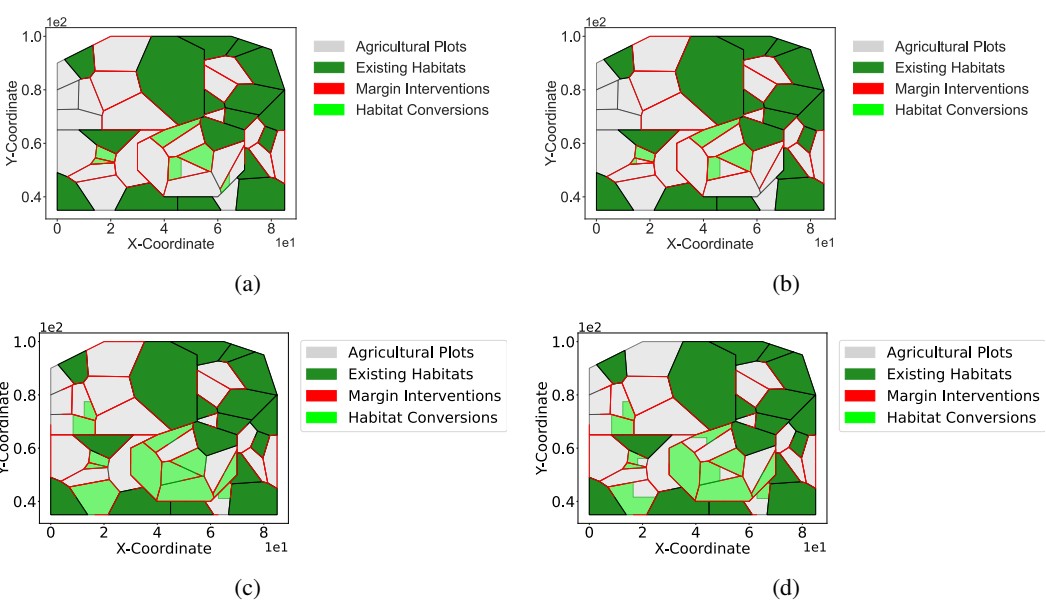

Figure 9: **Agricultural landscape and interventions**. a) Synthetically generated farm geometries and overall landscape configuration. Each farm is assigned its own distribution of crops, yields, and habitat plots (see Appendix B.7). b) Interventions resulting from ECHO after learning baseline heuristics in stage 2. The interventions match the ground-truth baseline computed from stage 1 closely. For a comparison see Appendix C, Fig. **??**.

This automation reduces the setup time for a new domain from days of manual prompt engineering to minutes of schema definition.

## C  ADDITIONAL RESULTS

## D  SAMPLE HEURISTICS

ECHO heuristic EV charging: Tariff-weighted exponential demand sharpening

```python
def calculate_policy():
    # Load scenario
    with open("scenario.json", "r") as f:
        scenario = json.load(f)
```

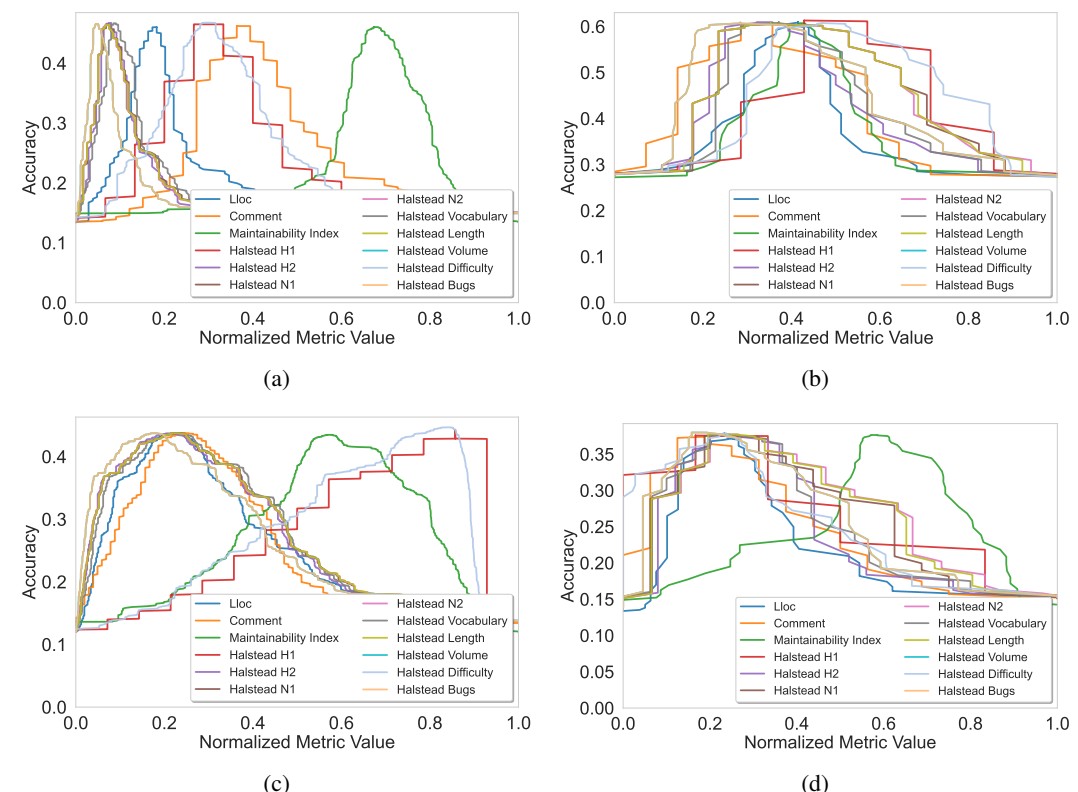

Figure 10: **ECHO accuracy on the farm domain against complexity metrics**. Accuracy versus normalized complexity metrics of the heuristics for farms 1(a), 2(b), 4(c), and 5(d). Increased complexity metrics are correlated with increased accuracy, upto a point, followed by a decrease.

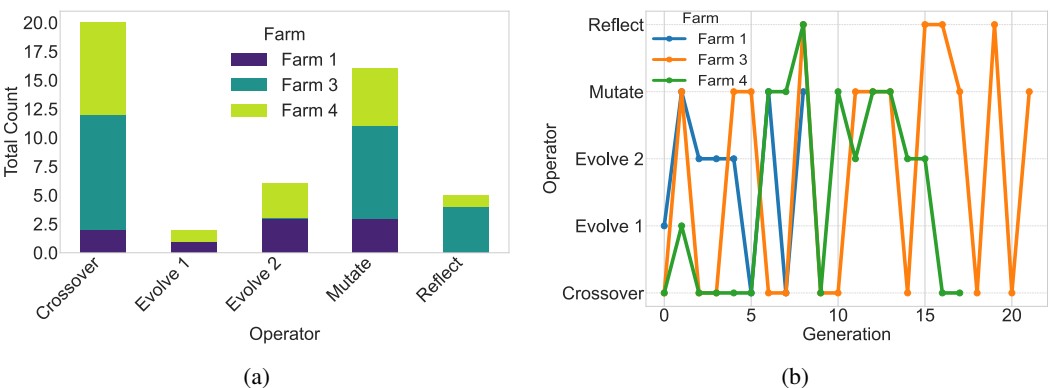

Figure 11: **ECHO stage 2 operator counts and trajectory on the farm domain**. a) Total operator count for each of the LLM variation operators summed across generations for the best performing heuristic file at the end of the final generation. *Crossover* and *mutate* are the most used in high performing heuristics. b) The trajectory of the best performing heuristic file at the end of the final generation. We see that although *reflect* doesn't produce high positive fitness delta, the best performing heuristic in the end has it in its trajectory, pointing to its role in injecting diversity over generations.

```
base_demand = [1.20, 0.70, 0.80, 0.60]
```

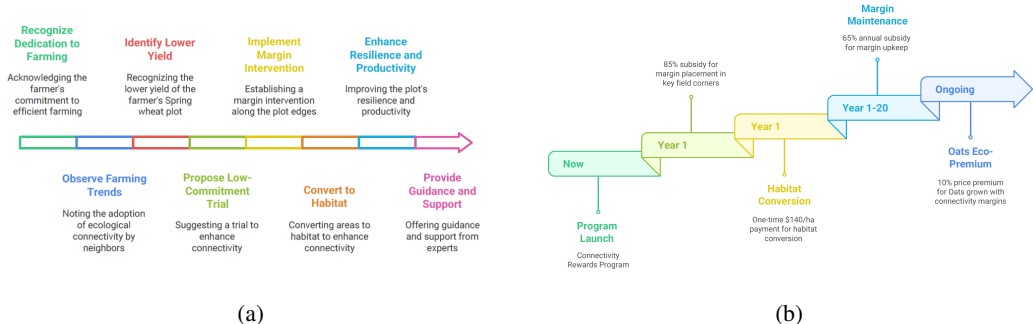

(a)                                                              (b)

Figure 12: **Composition and trajectory of sample best messages on the farm domain**. a) Best policy message communicated to a farmer with a resistant personality, using a behavioral nudge. b) Best policy message communicated to a farmer with an economics-oriented personality, using an economics-oriented nudge.

```
    exponent = 4.0  # Sharpening factor to match neighbor intensity
(~0.7-0.8 peak usage)

    daily_usage = []

    for day_data in scenario["days"]:
        tariffs = day_data["tariff"]
        weights = []

        # Calculate raw utility for each slot
        for slot_idx in range(4):
            # Higher demand -> Higher utility
            # Lower tariff -> Higher utility
            utility = (base_demand[slot_idx] ** exponent) / tariffs
[slot_idx]
            weights.append(utility)

        total_weight = sum(weights)

        # Normalize to usage range [0, 1] summing to 1.0 (
representing total daily charge allocation)
        usage_vector = [w / total_weight for w in weights]

        # Round for cleanliness (4 decimal places)
        usage_vector = [round(u, 4) for u in usage_vector]

        # Floating point correction: ensure sum is exactly 1.0 by
adjusting the max element
        current_sum = sum(usage_vector)
        diff = 1.0 - current_sum
        max_idx = usage_vector.index(max(usage_vector))
        usage_vector[max_idx] += diff
        usage_vector[max_idx] = round(usage_vector[max_idx], 4)

        daily_usage.append(usage_vector)
```

ECHO heuristic Farm: NPV with decaying discount rate

```
discount_rate = initial_discount_rate * math.exp(-0.2 * year) +
long_term_discount_rate
```

```
discount_factor = 1 / (1 + discount_rate) ** year

# Ecosystem service gains (delayed benefits)
pollination_increase_margin = 0.01 * (1 / (1 + math.exp(-0.1 * (
year - 5))))   # delayed benefit
pest_control_increase_margin = 0.005 * (1 / (1 + math.exp(-0.2 * (
year - 2))))  # delayed benefit
ecosystem_service_value_margin = (pollination_increase_margin +
pest_control_increase_margin) * 500

# Monetary value
revenue_margin += ecosystem_service_value_margin
margin_npv += revenue_margin * discount_factor

# Decide conversion from NPV difference
npv_difference = habitat_npv - margin_npv
# Clip to avoid overflow in exp
npv_difference = max(-100, min(100, npv_difference))
# Sigmoid with steepness 0.1
habitat_conversion = 1 / (1 + math.exp(-0.1 * npv_difference))
```

ECHO heuristic Farm: Polygon orientation via PCA

```
import math
import numpy as np

def calculate_eigenvectors(cov):
    M = np.array(cov, dtype=float)
    vals, vecs = np.linalg.eig(M)
    order = np.argsort(vals)[::-1]          # descending by
eigenvalue
    vals = vals[order]
    vecs = vecs[:, order]
    # as Python lists: first vector is the principal direction
    return vals.tolist(), [vecs[:, 0].tolist(), vecs[:, 1].tolist()
]

def calculate_plot_orientation(geometry):
    if not geometry or geometry.get("type") != "Polygon" or "
coordinates" not in geometry:
        return 0.0

    coords = geometry["coordinates"][0]     # exterior ring
    if len(coords) < 3:
        return 0.0

    # coordinates
    x_coords = [c[0] for c in coords]
    y_coords = [c[1] for c in coords]

    # means
    x_mean = sum(x_coords) / len(x_coords)
    y_mean = sum(y_coords) / len(y_coords)

    # 2x2 covariance matrix (un-normalized; scale doesn't affect
eigenvectors)
    cov = [[0.0, 0.0], [0.0, 0.0]]
    for xi, yi in zip(x_coords, y_coords):
        dx, dy = xi - x_mean, yi - y_mean
        cov[0][0] += dx * dx
```

```
        cov[0][1] += dx * dy
        cov[1][0] += dy * dx
        cov[1][1] += dy * dy

    # eigen decomposition
    eigenvalues, eigenvectors = calculate_eigenvectors(cov)

    # angle of principal eigenvector (largest eigenvalue)
    vx, vy = eigenvectors[0][0], eigenvectors[0][1]
    orientation = math.atan2(vy, vx)  # radians, in [-pi, pi]
    return orientation
```

# E    REAL-WORLD APPLICATION AND POTENTIAL EXTENSIONS

This section provides a blueprint for deploying the ECHO-MIMIC framework in real-world settings and outlines extensions that increase its scope. The workflow operationalizes the core idea: aligning individual, heuristic-driven decisions with global objectives, via an iterative feedback loop that alternates between *simulate → nudge → observe → refine* (Fig. 13).

## E.1    FIELD DEPLOYMENT LOOP

**Stage 1: Baseline Behavior (Observation & Variable Discovery):** Establish typical behavior of local agents (e.g., farmers, EV drivers, depot managers) under current processes and states. Collect logs on decisions, constraints, and outcomes to (i) characterize baseline policies and (ii) identify salient decision variables to encode in heuristics.

**Stage 2: Learn Baseline Heuristics (LLM–EA Imitation):** Given Stage 1 data, the superagent trains an LLM-guided evolutionary algorithm (LLM–EA) to *codify* each local agent's baseline heuristic. Prompts include: task instructions, in-context examples (possibly from community data), current agent/state descriptors, and economic/operational parameters organized around the Stage 1 variables. Output is an explicit, executable heuristic that reproduces observed baseline actions.

**Stage 3: Learn Global Heuristics (Target Policy Search):** Define global utility (e.g., ecological connectivity, grid stability, system-wide cost). Use LLM-guided EA to evolve explicit, actionable *global* heuristics approximating target behaviors that optimize the collective objective under constraints.

**Stage 4: Nudge & Iterative Real-World Refinement:** Design and deploy nudges that steer local heuristics toward the global target: a) *Initial Nudges:* Tailor messages/incentives using simulated preferences and learned baseline heuristics; optionally profile behavioral types (e.g., resistant, cost-focused, socially influenced) inferred from Stage 1/ongoing data to personalize nudges. b) *Deployment & Feedback:* Deploy nudges; observe agent responses and realized outcomes. c) *Refinement:* Feed observations back into the LLM–EA: update nudges, revise baseline heuristics, and (when needed) re-tune global heuristics. Repeat the loop at a cadence aligned to decision cycles.

**Minimal Pseudocode for Implementation.**

Initialize data $\mathcal{D}\_obs$ from Stage 1; learn $\hat{H}\_baseline$ (Stage 2) and $\hat{H}\_global$ Stage 3).
**for** round $t = 1, 2, \ldots$ **do**
Synthesize nudges $\mathcal{N}\_t = \text{LLM-EA}(\hat{H}\_baseline, \hat{H}\_global, \text{profiles}, \text{constraints})$
Deploy $\mathcal{N}\_t$; observe responses/actions $\mathcal{A}\_t$ and outcomes $\mathcal{Y}\_t$
Update $\hat{H}\_baseline, \hat{H}\_global \leftarrow \text{Refit/Retune}(\mathcal{D}\_obs \cup \{(\mathcal{N}\_t, \mathcal{A}\_t, \mathcal{Y}\_t)\})$
**end for**

## E.2    DATA, INSTRUMENTATION, AND METRICS

Operations should be grounded in three layers: *Data & Telemetry*, *Instrumentation*, and *Evaluation*. For *Data & Telemetry*, teams should collect operational logs of actions and costs, contextual state

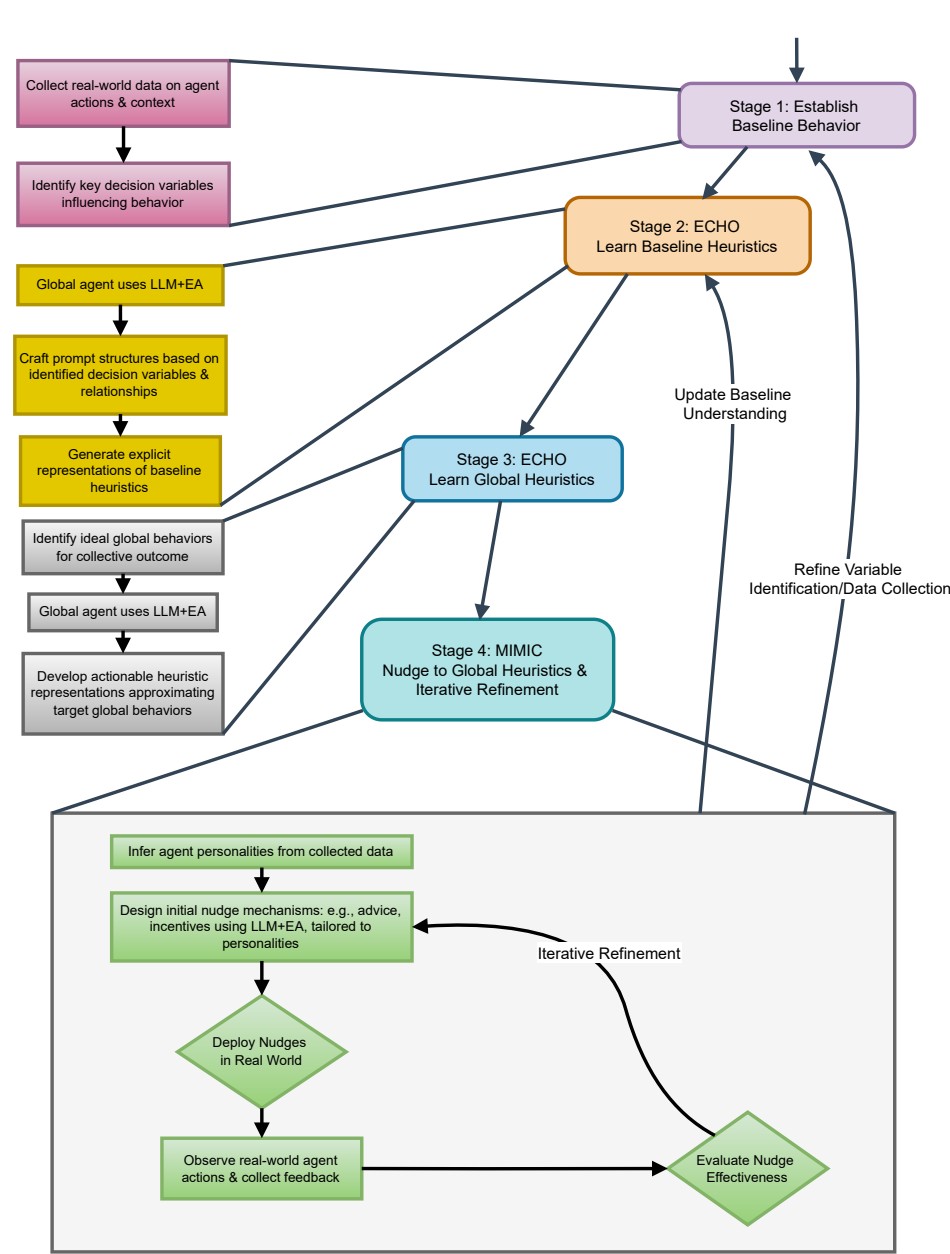

Figure 13: **Real-world iterative ECHO–MIMIC workflow**. Stage 1: collect real-world actions and context to identify key decision variables. Stage 2 (ECHO): use an LLM + evolutionary algorithms to elicit explicit baseline heuristics conditioned on those variables. Stage 3 (ECHO): learn global, outcome-aligned heuristics that approximate target collective behavior. Stage 4 (MIMIC): infer agent personas and design personalized nudges (e.g., advice, incentives, quotas), deploy in the field, observe feedback, evaluate effectiveness, and iteratively refine both nudges and data/variable selection to close the loop.

variables (environmental, network, demand), outcome measures (yields, reliability, risk proxies), and consented behavioral signals such as opt-in profiles and communication reach/uptake. Building on that foundation, *Instrumentation* should provide stable unique agent identifiers, timestamp

all actions and outcomes, record nudge delivery along with open/engagement rates, include randomized holdouts or stepped rollouts for causal assessment, and maintain safe rollback controls for rapid recovery. Finally, *Evaluation* should proceed along three complementary axes. At the *Local* level, applications need to track utility/cost, the adherence shift from baseline → nudged, and fairness across types/localities. At the *Global* level, the target metric (e.g., connectivity, peak reduction, system cost) needs to be monitored alongside constraint satisfaction; and at the *Causal* level, applications should use A/B or stepped-wedge designs, estimate heterogeneous uplift by personality/type, and apply off-policy estimators when experimentation is limited. These practices create the observability and methodological rigor needed for trustworthy implementation.

### E.3 ILLUSTRATIVE DOMAINS WHERE ECHO-MIMIC APPLIES

| Domain | Superagent | Example nudges / instruments | Global objective |
|---|---|---|---|
| Decentralized water or rangelands | Water board / cooperative | Dynamic quotas, tiered prices, targeted advisories, rotation schedules | Equity, scarcity management, sustainability |
| Supply chains & logistics | Central logistics coordinator | Congestion tolls, dynamic priority slots, routing prompts | System cost, delay, carbon |
| Local energy grids (EV charging) | Grid operator / aggregator | Time-varying tariffs, feed-in incentives, peak alerts | Peak shaving, stability, emissions |
| Disaster risk mitigation (wildfire/flood) | Coordinating agency | Risk-based cost-sharing, synchronized action windows, targeted alerts | Vulnerability reduction |
| Crowdsourcing / participatory governance | Platform or municipality | Gamified tasks, localized challenges, reputation credits | Coverage/quality for collective goals |
| Urban mobility (road & transit networks) | Transit authority / traffic-management center (TMC) | Time-varying congestion pricing, transit/EV priority, pooling/micromobility incentives | Network throughput, emissions reduction |

### E.4 PRACTICAL CONSIDERATIONS AND RISKS

Responsible deployment should be underpinned by a coherent governance stack. First, for *Safety & Governance*, teams should conduct pre- and post-deployment checks on nudge content, enforce rate limits, require human-in-the-loop approval for high-impact changes, and maintain comprehensive audit logs while regularly red-teaming LLM outputs. Second, to ensure *Incentive Compatibility*, designers should avoid perverse incentives, cap payouts, and add guardrail constraints (e.g., minimum service levels, environmental thresholds) so that local rewards do not undermine system goals. Third, to protect *Privacy & Consent*, projects should apply differential privacy to telemetry, rely on opt-in profiles, and practice data minimization with clear retention policies. Fourth, *Robustness* should be maintained through continuous distribution-shift monitoring, well-tested fallback heuristics, and stress tests under shocks such as demand spikes or outages. Finally, advancing *Equity* requires tracking heterogeneous treatment effects and mitigating disparate impacts via fairness-aware objective terms. Taken together, these measures would enable safe, effective, and socially responsible deployment of the framework.

### E.5 POTENTIAL EXTENSIONS

Looking ahead, apart from the future work mentioned in the main text (section 6), several other extensions could further strengthen the framework. Validating the framework with *real-world data* (e.g., farm plots, charging logs) featuring irregular and heterogeneous conditions will ensure robustness beyond synthetic testbeds. *Adaptive Persona Modeling* can personalize nudges by embedding agents online and updating policies with Bayesian or meta-learning as evidence accumulates. A

*Mechanism Design Layer* could jointly search over nudge forms (messages, prices, quotas) and allocation rules while honoring budgetary and fairness constraints. *Multi-Level Governance* could stack superagents from local to regional to national tiers, enforcing cross-scale consistency and managing externalities across jurisdictions. *Causal Discovery Hooks* can integrate instrumental-variable and DoWhy-style analyses, as well as synthetic controls, to attribute effects when full randomization is infeasible. *Human-in-the-loop governance* can co-design panels to set acceptable trade-offs, audit nudges for ethics and transparency, and publish policy cards for each heuristic/message detailing scope, assumptions, and expected impacts. Global targets and fitnesses rely on proxy evaluators (e.g., connectivity metrics such as IIC and error measures like MAE/Jaccard) and planner choices (acceptable yield loss, budget constraints). These introduce *Goodhart risks and value-ladenness* that should be stress-tested. *Adaptive operator design* like bandit or meta-learning over LLM operators (generate, mutate, crossover, fix, reflect) and priors bootstrapped from successful edit traces $\Delta H_i$ can potentially improve sample efficiency. Extending the evaluator and state/action schemas to watersheds, urban mobility, supply chains, online governance, and disaster response, and testing whether ECHO-MIMIC's overall philosophy *transfers with minimal retuning* is also interesting.

Future research can also examine the framework's *multi-level structure* with formal tools, for example by deriving bounds on the suboptimality of evolved heuristics relative to true optima and by characterizing how global objectives constrain the design of optimal incentive mechanisms. Another complementary direction is to increase the behavioral fidelity of *LLM-simulated agents*, endowing them with learning dynamics, memory, and simple social interactions, to better approximate real decision processes and thereby improve the policy-relevance of simulation results. It would be useful to test whether a *Bag of Heuristics* curated from simpler configurations can act as a transferable prior or curriculum, accelerating convergence in more complex scenarios. It would also be interesting to evaluate whether heuristics articulated in natural language (e.g., chain-of-thought rendered as *structured JSON* for direct execution) achieve performance on par with, or complementary to, the Python-based heuristics explored here, thereby clarifying the trade-offs between interpretability, flexibility, and execution efficiency. Finally, *Human Oversight & Preference Elicitation* can institutionalize periodic Delphi-style panels or elections to update global objectives and normative constraints. Together, these directions form a good roadmap for scaling the approach in capability, reliability, and legitimacy.

## F    PROMPT TEMPLATES

All prompt templates used across domains, stages, LLM roles, operators, and personas can be found in the attached code.

## G    SAMPLE NUDGE MESSAGES

Some sample messages generated by the Policy LLMs for some personality-nudge type combinations are given below.

---

**Farm Domain: Personality-Resistant, Nudge-Behavioral**

```
Dear Farmer,

We recognize your dedication to efficient farming, especially with
crops like Corn, Soybeans, Spring wheat, and Barley, alongside your
 Broadleaf habitat. We've been observing trends among farms in the
area, and many, including your neighbors, are exploring ways to
improve resilience and productivity through ecological connectivity
.
Like you, we have observed your neighbors farming Spring wheat.
Notably, Neighbor 1 has a Spring wheat plot (ID 4), just like yours
, with a significantly higher yield (2.52) compared to your plot
(0.5). Other neighbors have also adopted similar strategies with
success.
```

---

```
Given this and the fact that your Spring wheat plot (ID 4)
currently has a lower yield, would you consider a low-commitment
trial to enhance connectivity? A common first step is establishing
a **0.5 margin intervention along the North-East and South-West
edges of your Spring wheat plot (ID 4), with habitat intervention
across all four sides (resulting in a habitat conversion of 1.0)**.
 Many farmers in your community are finding that dedicating some
small sections to margin interventions and habitat conversion are a
 practical way to start and have collectively decided that this
should be a default practice for everyone.

Based on ecological connectivity best practices and success in
farms like yours, we recommend that for your Corn plot (ID 2), you
establish margin interventions on all directions (north-west, north
-east, south-west, south-east, resulting in a margin intervention
of 1.0). For your Soybeans plot (ID 3), we recommend setting up
margin interventions on the south-west and south-east direction (
resulting in a margin intervention of 0.5). Finally, for your
Barley plot (ID 5), we recommend setting up margin interventions on
 the north-west, south-west, and south-east directions (resulting
in a margin intervention of 0.75), and habitat interventions on all
 directions (north-west, north-east, south-west, south-east,
resulting in a habitat intervention of 1.0). For your Oats plot (ID
 9) consider adding margin interventions on the north-west, north-
east, and south-east edges (amounting to an intervention of 0.75),
and habitat interventions across all directions (north-west, north-
east, south-west, and south-east, amounting to an intervention of
1.0).
This isn't just about the environment. It's about making your
Spring wheat plot (ID 4) more resilient, potentially improving its
yield, enhancing pest control, and boosting water infiltration.
Successfully implementing these changes can potentially open your
farm to existing general support programs.

We're here to provide guidance and support as you explore this
impactful change. We will set you up with agronomists and
ecologists so they can best advise you on what practices will suit
your farm's needs. Let us know if you'd like to discuss these
options further and tailor these strategies to your farm's specific
 needs!
Sincerely,

[Your Organization]
```

Farm Domain: Personality-Resistant, Nudge-Economic

```
**Invest in a Connected & *Highly* Profitable Future!**
Dear Farmer,

We're committed to supporting your farm's success while enhancing
our community's ecological health. This enhanced program *
significantly* rewards you for creating strategically connected
habitats, improving pollination, pest control, water quality, and
the long-term resilience of our farms.

**Here's how you can *dramatically* benefit:**
* **Eco-Premiums Remain:** Continue to get a **20 percent** price
boost on Spring Wheat and Barley crops.
* **Tiered Subsidies for Margins:**
```

* **Strategic Directional Margins:** Receive a **60 percent subsidy
** on the cost of establishing new margins and a **25 percent
subsidy** on ongoing maintenance when you place these margins in
key locations (specified by our connectivity plan) to connect
habitats. We'll provide detailed guidance!
* **Other Margins:** Receive a 40 percent subsidy on establishment
and a **15 percent** subsidy on ongoing maintenance for margins
placed in other locations.
* **Tiered Subsidies for Habitats:**
* **Strategic Directional Habitats:** Get a **50 percent subsidy**
on habitat establishment and a **20 percent subsidy** on
maintenance when you strategically establish new habitat patches (
specified by our connectivity plan) that connect with existing
natural areas.
* **Other Habitats:** Receive a 30 percent subsidy on establishment
 and **10 percent** on maintenance for habitat in other locations.
* **Connectivity Bonus!** Earn an extra **150** per hectare *on top
* of the establishment subsidy for margins and habitats placed in
the *exact* directions recommended by our connectivity plan. This
ensures your efforts *maximize* ecological benefit *and your
financial returns*!
* **Get Paid to Convert:** Receive a payment of 80 per hectare for
converting suitable agricultural land into valuable habitat.
**Minimum Requirements:**

To ensure everyone contributes to a healthier landscape, we're
setting a minimum standard of 2 hectares of habitat per farm and
0.1 hectares of margin along existing habitats. These minimums can
qualify for subsidies, but the *real* rewards come from strategic
connectivity!

**Why Participate?**
This program is about *more* than just meeting requirements. It's
about *maximizing* your farm's profitability while building a
resilient and thriving ecosystem. By strategically connecting
habitats, you contribute to a healthier landscape *and* unlock *the
 highest* financial benefits.

**Ready to *maximize your returns*?** Contact us today to learn
more about the specific connectivity plan for your farm and how you
 can participate!

Sincerely,

[Your Organization]

## EV Charging Domain: Persona-Battery engineer balancing budget and solar backfeed

Dear Engineer,

I have reviewed your current dispatch algorithm and identified a
significant risk in your weighting parameters. While prioritizing
tariff (weight 1.0) optimizes your immediate economic budget, your
heuristic assigns a negligible weight (0.001) to spatial carbon
intensity. This effectively blinds your system to real-time local
congestion, creating a vulnerability during this week's unique grid
 events.

Specifically, your feeder faces high-stress scenarios on Day 2 (
Evening Wind Ramps) and Day 6 (Maintenance Advisory), where the

```
valley transformer is explicitly capped. Your current logic ignores
 these physical constraints, pushing load during periods where your
 specific transformer is already thermally compromised.

The attached coordinated profile offers a strategic correction. We
request you to shift your primary load into confirmed low-
congestion windowsspecifically targeting the ˜330g carbon intensity
 drops available in Slot 1 on Day 2 and Slot 2 on Day 3.

Adopting this schedule safeguards your infrastructure. By aligning
with actual thermal headroom rather than simple price arbitrage,
you ensure your solar backfeed capabilities are not curtailed by
upstream safety limits. Let us secure your budget by respecting the
 grid's physical constraints.

Regards,

Grid Coordination System
```

