# OpenReview forum: "Structuring Collective Action with LLM-Guided Evolution: From Ill-Structured Problems to Executable Heuristics"
_ICLR.cc/2026/Conference — Submitted to ICLR 2026_

### Official Review · Reviewer_mHoj · 2025-10-28

**Soundness:** 3
**Presentation:** 3
**Contribution:** 2
**Rating:** 2
**Confidence:** 3

**Summary:**

This paper applies the idea of Evolutionary Search with LLMs to the problems of (1) navigating an ill-structured collective action dilemma (agricultural landscape management) and (2) finding appropriate heuristics to resolve the dilemma. Specifically, they introduce the ECHO-MIMIC method where the ECHO component comes up with creative but useful heuristics and nudges, the MIMIC part then tries to convince an agent to use them.

**Strengths:**

- The paper seems original in the sense that I haven't seen an LLM-driven evolutionary search approach being taking for any collective action problem. I get the impression that proper effort has been put into getting a good design of their method ECHO-MIMIC. The paper writing was clear.
- I like that this paper chose to a test case that, from a first glance, seems to capture much of the complexity of a real-world collective action problem

**Weaknesses:**

Most of my overall negative assessment of this paper is based on the fact that I do not view this contribution as significant enough to the ICLR community. I am interested to hear from the AC and other reviewers how they feel about it. I will elaborate on two complementary fronts on which I feel like the paper has not demonstrated yet that their main contribution, the method ECHO-MIMIC, is effective.
- What evidence do we have that a capable LLM agent, say in two years time, is not able to surpass the performance of ECHO-MIMIC with the newest LLM at that time? ECHO-MIMIC is introducing a lot of constraining structure onto the problem of solving the agricultural landscape management issue. They present DSPy MiPROv2 as "a strong LLM-native baseline", and show that ECHO-MIMIC performs much better. But I yet have to feel convinced that a framework like ECHO-MIMIC can hold its ground against increasingly capable general LLM agent scaffolds that can be prompted to resolve the collective action problem. I suggest adding a side by side comparisons of how ECHO-MIMIC scales with stronger base LLMs vs how a general LLM agent framework scales with stronger base LLMs.
- While the agricultural landscape management is interesting to our society in its own right, I do not think that it is sufficiently interesting to the ICLR community to serve as the only test bed for your method. It is hard to gauge how generalizable your insights about ECHO-MIMIC are *beyond* your particular problem of interest (say, other collective action problems or social dilemmas). To me, the current narrative and focus of the paper seems to fit much better into non-ML venues with a stronger emphasis on solving individual collective action problems. (This is despite acknowledging that your method ECHO-MIMIC is fully ML-based.)
- Less important, but maybe also illustrating the point above: The paper did not put much effort into putting their work in the context of related ML/AI work that is addressing (ecological) collective action problems / social dilemmas.

**Questions:**

When you decided to tackle the ill structured problem through a dual approach on the agent level as well as on the global policy maker level, what factors drove your decision to introduce a policy level agent specifically with regards to how much power you give it for influencing the individual agents? I am asking this because there are many ways to resolve a social dilemma from the perspective of a policy maker with different powers; for example, through reputation systems, contracts, mediation, etc.

---

> ### Author Response · Authors · 2025-11-22
> **Demonstrating Generalization and Relevance to ICLR Community**
>
> We thank the reviewer for noticing the strengths in our paper and providing constructive comments. We address your comments in order below.
>
> 1. "Can ECHO-MIMIC hold its ground against increasingly capable general LLM agent scaffolds that can be prompted to resolve the collective action problem"
>
> We thank the reviewer for their comment. In general, we prescribe to the reviewer’s philosophy that we shouldn’t be wasting time designing things that immediate future general purpose LLM agents can one shot. We say immediate future here because this might still be worth doing if that general purpose LLM agent is in the far future. We also agree using unnecessary constraining structure in general is bad as evidenced by the times we live in. However, this line of thinking may be misplaced here.
>
> Asking a question of the form “if a capable future LLM agent scaffold is able to come close to solving collective action, why bother designing explicit frameworks that achieve it” is like asking “if a capable future LLM agent scaffold is able to come up with mathematical breakthroughs, why bother designing something like AlphaEvolve”. We are not claiming here that our work is as significant as AlphaEvolve, but just pointing out that this line of thinking may not be productive. If a future general purpose LLM agent scaffold does indeed perform better at collective action problems (after problem specific set up or tuning), we argue that its scaffolding would borrow from the way we have designed our framework and its stages.
>
> Collective action is not a problem that a LLM agent can one shot, unlike writing code or even proving new theorems. It’s a system level problem that requires breaking down the problem, solving it at different levels, and then combining it in a meaningful way. Even then, we are not close to solving collective action in any meaningful sense. The real-world implementation constraints are gargantuan. Our approach here just provides an initial framework towards thinking about this problem, using LLMs and evolutionary algorithms for their strengths (heuristic program discovery and automated persuasion) and making up for their weaknesses by explicit multi-stage structural scaffold. Can a future LLM completely get rid of the evolutionary process and one shot heuristic program discovery, thereby rendering part of our pipeline irrelevant? It's possible that something like RL alone or a future technique can get us to superhuman program discovery. Does it mean we stop using EA’s to find better task specific programs today? Probably not.
>
> So what we are saying is, you may be right that a future scaffold may be better at solving these kinds of problems, but there is a high chance that scaffold looks similar to ours in which case it would have drawn inspiration from us. There is a chance that the scaffold doesn’t look like ours at all, in which case it would have solved issues in our current version, and therefore drawn inspiration from us. The fact that we are one of the first to think about this problem this way, and demonstrate it in a close to real world setting, we believe, deserves some credit.
>
> As for a scaffold that is fully general purpose (without any problem specific setup or tuning) to solve these kinds of system level problems, essentially it would have to create the entire ECHO-MIMIC workflow (or something analogous) from scratch. This is definitely possible and may even be in the immediate future, we can’t rule it out completely. However, thinking about difficult problems with the tools available at our disposal is what we like to do, and we will continue to do so without waiting for a powerful future AI to do it for us. We will also continue to do so even when the future AI is among us, only now our tools would include it.
>
> Regarding your comment about comparing how ECHO-MIMIC and a general purpose LLM agent framework scale with stronger base LLMs, we agree that we didn’t do a good job of comparing our method to existing scaffolds. We have now included a comparison with a general purpose agent scaffold (**AutoGen**) and included details about how both scale with *stronger LLMs* (Gemini 2.5 Pro and GPT-5 mini). See Table 1. Wee see that though baselines perform better with more capable LLMs (G2.5-P, GPT5-m), their performance cannot match ECHO-MIMIC, which also benefits from higher capability.

---

> ### Author Response · Authors · 2025-11-22
> **Continuing previous comment**
>
> 2. "Relevance to ICLR"
>
> We agree that agricultural landscape management alone is not as interesting to the AI/ML/ICLR community. We chose this domain because we had prior experience thinking about the real world details of this domain and wanted to make our demonstration task as close to the real world as possible. However, we agree it cannot serve as the only testbed for our method as we have not demonstrated generalization to other domains.
>
> We have been working on integrating more domains into our pipeline since submission, and we have now added a second domain, **Carbon-Aware EV Charging Slot Choice**, with different topology (simpler stylized geometry) and goals. We modify the (state, action) schema, evaluator, observability, topology, and context, and re-use the same ECHO-MIMIC stages. The relative gains over the added baselines persist, supporting domain portability and generalization. A brief introduction to the domain along with the performance is given in the results, and more details about the domain and task is provided in the appendix.
>
> We have now also created a **Domain Creation Agent** that automatically writes instructions and evaluations in the desired modular format given the (state, action) schema of agents, observability constraints, type of ground truth information, background context, and additional relevant information of the domain specific task at hand. It helps generate system instructions, task prompts, and evaluation harness for the ECHO and MIMIC stages, automatically adapting to the specific terminology and logic of the new domain, allowing our framework to scale to new collective action problems.
>
> We elaborate more in the paper about how given the required information about the task, our custom Domain Creation Agent, modular and composable instruction format, and the ECHO-MIMIC pipeline overall, our approach generalizes across domains as demonstrated. We believe that this makes ECHO-MIMIC a powerful way to think about collective action problems, and a novel and important use of LLMs under the field of AI for public/social good. Therefore, we believe ICLR is the right venue for this contribution.
>
> 3. "ML/AI work that is addressing (ecological) collective action problems/social dilemmas"
>
> We agree that we should have done a better job of discussing previous ML/AI work that addresses collective action and the like, and how it relates to our work. We had conducted this literature search, but it got left out of the submitted version. We have now added a separate subsection for this in the *Related Work* section.
>
> The short summary of the section we added is that there is ML/AI work on *sequential social dilemmas* and *inequity aversion*, and some others, but most of these use stylized spatial geometries (gridworlds, local observation windows) that are useful as testbeds but do not target a domain-specific deployment. Our work on the other hand is demonstrated in both a real-world spatial deployment and a stylized geometry (newly added domain of EV Charging). Our work is complementary to existing work and closer to practice. Rather than optimizing opaque neural policies or only reshaping rewards, we evolve executable local programs (closer to how humans actually make decisions). Prior works optimize gradients, utilities, or communication in synthetic arenas, whereas we design an end-to-end framework to think about collective action in general, and demonstrate it in two different settings by producing deployable rulebooks plus matching messages.
>
> Q1. "how much power you give the global policy maker for influencing the individual agents"
>
> We deliberately began with *low-power, consent-based* influence at the policy level so we could identify the causal effect of an intervention (message or incentive) without confounding it with enforcement. Agents make a single decision. Adding coercive tools (penalties, binding constraints, repeated monitoring) would blur whether improvements came from better reasoning or from compulsion. Low-power instruments also keep the learning signal clean (accept/refuse leading to concrete code edit or not) and preserve interpretability (what changed and why).
>
> Although ECHO-MIMIC uses an evolutionary loop with many training iterations, these are optimizer rounds in policy/message space, not bargaining with an agent. The optimizer proposes candidates, evaluates whether the farm agent edits its heuristic code and the resulting global score, then selects/mutates. At deployment, interaction is one-shot: the policy agent sends one consent-based message (optionally with a budget-feasible incentive offer). The farm accepts or refuses and updates its heuristic once. There is no sequential negotiation, renegotiation, dynamic penalties, or reputation play.

---

> ### Author Response · Authors · 2025-11-22
> **Continuing previous comment**
>
> Our current version already spans two meaningful rungs on the policy-maker “power ladder”:
>
> a) *Behavioral-economics levers* (consent-based, zero-coercion): Our policy agent crafts an agent-specific message using (i) social comparisons to nearby peers, (ii) defaults that make a recommended “first step” salient, (iii) lightweight commitments that the agent can voluntarily adopt, and (iv) careful framing of global outcome benefits and concerns. The message is instantiated from the agent’s own and neighbors’ data and evaluated by checking whether the *Agent LLM* edits its heuristic code from local heuristics to global heuristics or refuse?
>
> b) *Bounded economic incentives* (price-based, still voluntary): The policy agent may also propose a budget-compliant package with explicit rates within fixed bounds, subsidy factors and one-time payment, subject to a strict budget. Again, adoption is voluntary.
>
> This behavioral + bounded-incentive design addresses your question about how much “power” the policy agent has. We give it choice architecture and priced offers, but no hard constraints, penalties, or reputation loss, and no multi-round bargaining. We did this to: (i) isolate the mechanism’s effect in a one-shot decision; (ii) keep the evolved policies and messages trackable; and (iii) avoid turning the paper into a many-round mechanism-design study with additional assumptions about monitoring and compliance.
>
> That said, our pipeline naturally extends to richer institutions you mention, using the same “message to code-change (or refusal)” evaluation:
>
> a) *Reputation systems*: Add an agent-level “reputation/standing” state and a policy-side subroutine that posts reputation updates contingent on the adopted action. The message then includes the prospective reputation effect; adoption is still voluntary. Reputation would enter the agent’s local utility (e.g., access to programs or preferred pricing).
>
> b) *Lightweight contracts/pledges*: Allow the policy agent to propose a voluntary contract: a simple commitment (e.g., “do X for Y years with Z% subsidy; exit penalty = 0”). Technically this is just a richer message that, if accepted, adds a clause to the agent’s heuristic.
>
> c) *Mediation/coordination with side-payments*: In a one-shot setting we can still prototype a mediated joint plan: the policy agent proposes a coordinated action profile plus budget-feasible side-payments (our current incentive instruments already support this), and each agent adopts the corresponding local rule if the offer meets its guardrails. This preserves consent while exploring stronger forms of centralized coordination.
>
> Practically, these extensions require only small schema changes (extra state for reputation/pledges), plus clear budget and fairness constraints for any transfers. The ECHO-MIMIC stages are unchanged. Our current implementation already exercises two policy modalities (behavioral and economic) in a principled, budgeted way, and the same message to code-edit machinery lets us incorporate reputation, simple contracts, or mediated side-payments in future work without abandoning the one-shot, consent-based philosophy.
>
> In summary, we added  an additional baseline (AutoGen), comparison with stronger base LLMs (Gemini 2.5 Pro and GPT-5 mini), a second domain (Carbon-Aware EV Charging Slot Choice), a Domain Creation Agent, and demonstrated generalization, expanded literature review, and clarified novelty. Apart from our discussion, we also added an additional LLM family (GPT-5 nano), expanded the nudging section, improved  the figures, and clarified the core idea. We hope that these changes have made our paper much stronger than the previous version, and we respectfully ask you to reconsider your rating for our paper.

---

### Official Review · Reviewer_smN3 · 2025-10-30

**Soundness:** 3
**Presentation:** 2
**Contribution:** 3
**Rating:** 6
**Confidence:** 3

**Summary:**

The authors present ECHO-MIMIC, a two-stage computational framework that transforms collective action problems, framed as Ill-Structured Problems (ISPs), into Well-Structured Problems (WSPs) for individual agents.
ECHO (Evolutionary Crafting of Heuristics from Outcomes) evolves executable heuristics, snippets of code encoding behavioural policies, while MIMIC (Mechanism Inference & Messaging for Individual-to-Collective Alignment) generates persuasive natural-language messages that motivate agents to adopt those heuristics. Both modules use large-language-model-guided evolutionary search and population-level selection within a simulated environment to optimise for global performance.
Demonstrated on an agricultural landscape management problem, ECHO-MIMIC discovers effective local heuristics and accompanying messages that align farmer behaviour with landscape-level ecological goals. This work reframes the challenge of collective action as an agent-level policy discovery and communication problem, offering a scalable route to adaptive policy design.

**Strengths:**

- Novel approach using evolutionary algorithms and LLMs to adaptively design policies for decision-making
- Interesting idea on the "nudging" mechanism

**Weaknesses:**

- End-to-end example prompt & completion outputs would be helpful in the main body of the paper and help clarify the core ideas and hypothesis,
- I would be interested in more detailed resutls and discussion on the "nudging" mechanism.
- Diagrams could be more legible and help the reader understand the projects much quicker.
- I gave a "fair" for Presentation because I believe the clarity and flow of the paper could be improved significantly

**Questions:**

- Can agents refuse "Nudge Messages"?
- On high-level afaik, nudge messages are compiled using collective local information, however I wonder if hirarchical or some parts of the nudge message might be more important than others depending on the local condition. For example, the "message" i receive from my direct neighbour might be more helpful to take action, the "message" from a farmer on the other side of the board not so much.

---

> ### Author Response · Authors · 2025-11-22
> **Clarifying Workflow, Expanding Nudging Mechanism, and New Domain**
>
> We thank the reviewer for noticing the strengths in our paper and providing constructive comments. We address your comments in order below.
>
> 1. "End-to-end example prompt & completion outputs"
>
> We thank the reviewer for their comment. We agree that the core idea of the paper can be presented better, however, we think including an end to end example with full prompt completion outputs in the main body is probably not feasible. We dedicated entire sections in the appendix with multiple pages for this, for example Appendix D “Sample Heuristics” showing some outputs from Stages 2 and 3, Appendix F “Prompt Templates” showing all the detailed prompts at all stages, and Appendix G “Sample Nudge Messages” showing some outputs from Stage 4. In the main body, we stick to presenting only the templates and composition of prompts, and how all the stages fit together.
>
> We also mention in the caption of Figure 1 that for a more detailed workflow, readers can refer to Appendix B.2. In Appendix B.2, in Figure 5 and 6, we lay out the entire workflow step by step in flowchart form, including the different subroutines and LLMs used. This combined with the overall idea presented in Figure 1, the prompt compositions and details mentioned in Section 4 “The ECHO-MIMIC Framework”, The prompts and outputs given in Appendices D, F, G, and source code, and details given about “Real-world Applications and Potential Extensions” in Appendix E, we believe conveys the main idea about what we are trying to achieve. Therefore, we omit this due to space constraints.
>
> 2. "Detailed results and discussion on the nudging mechanism"
>
> We agree that the nudging mechanism section can use more detail. We now discuss more about how agents can refuse to change, and who actually nudges. We already discuss in the paper about how the prompts are constructed for each persona and what kind of information is used to construct the nudge messages when we talk about the different LLMs used in MIMIC in section 4.2. For a more detailed workflow of MIMIC with a step by step flowchart, we point you (and the reader) to Appendix B.2; Fig. 5b, and for full best-message exemplars to Appendix G.
>
> We now add a result for the score trajectory over generations for nudging in Stage 4 for a second domain (Carbon aware EV charging). See Fig. 4a. We see that in the EV charging domain, with versatile personas (to model agent heterogeneity) and generic instructions (no specific framing), accuracy with respect to generated global heuristic actions from ECHO (stage 3) improves across generations and agents. MIMIC learns to use personalized benefits, social proof, and environmental impact framing at different points in the trajectory. Overall, the experiments in 4a and 4b both demonstrate the persona and framing specific targeting potential of MIMIC. For qualitative examples, refer to Appendix G.
>
> For a even more detailed description about the nudging mechanism, please refer to response to Q1: "how much power you give the global policy maker for influencing the individual agents" by Reviewer mHoj.
>
> 3. "Diagrams could be more legible"
>
> We agree that the diagrams can be improved further and we have done our best now to fix them. Is there any particular diagram that you think needs improvement? As all diagrams are vector images, they can be zoomed considerably for clarity. Do you mean the fonts can be larger? The line strokes can be thicker? Would be helpful if you can elaborate.
>
> 4. "Clarity and flow of the paper could be improved significantly"
>
> We agree that the flow sometimes seems jagged. We have now tried to clarify the core idea of the general end-to-end framework for collective action. We have also clarified the inputs/outputs of all stages, removed unnecessary pieces of text, improved and rearranged the diagrams to the best of our ability, demonstrated generalization by adding a second domain of EV charging, and expanded on the nudging mechanism section to include more details. We hope that these changes have improved the clarity and flow of the paper.
>
> Q1. "Can agents refuse Nudge Messages"
>
> Yes, *agents can refuse nudge messages*. There is an explicit instruction that tells the agents that they can and they should refuse to change their behavior unless they believe the nudge messages truly benefit them and their persona. This mimics the real world, where people mostly don't change their behaviour as a default, unless they are persuaded using mechanisms that either benefit their personal situation explicitly or cater to their emotional and social needs/principles.

---

> ### Author Response · Authors · 2025-11-22
> **Continuing previous comment**
>
> Q2. "nudge messages compilation and communication"
>
> We agree regarding your intuition that messages and information from close by neighbours is more relevant for behavioural change than from further away. However, in our setting, the nudge messages are not sent by other agents but by the *central coordinator*. The central agent looks at the local heuristics of agent i (generated from stage 2) and the potential global heuristics of agent i (generated from stage 3), and then comes up with nudge messages that move the behavior of agent i from the former to the latter. It is true that locally proximal information is more useful and relevant in general, however, this is already baked into our stage 3 heuristics, which try to approximate the global ground truth. When coming up with heuristics for stage 3, the locally proximal information is what is fed into the LLM (to mirror societal information diffusion). This combined with the fact that the ground truth metric is influenced by local geometry and inputs, already makes what you said true in our framework.
>
> In summary, we expanded the nudging section and clarified the core idea of our work. Apart from our discussion, we also added an additional baseline (AutoGen), comparison with stronger base LLMs (Gemini 2.5 Pro and GPT-5 mini), a second domain (Carbon-Aware EV Charging Slot Choice), a Domain Creation Agent, and an additional LLM family (GPT-5 nano). We also demonstrated generalization, expanded literature review, and clarified novelty. We hope that these changes have made our paper stronger than the previous version, and we respectfully ask you to reconsider your rating for our paper.

---

### Official Review · Reviewer_dnJu · 2025-10-31

**Soundness:** 3
**Presentation:** 4
**Contribution:** 3
**Rating:** 8
**Confidence:** 3

**Summary:**

This paper proposes ECHO-MIMIC, a two‑phase LLM‑guided evolutionary framework that turns messy collective‑action problems into executable local rules and matching persuasive messages. ECHO evolves short Python programs that map each agent’s local state to actions: first to imitate profit‑seeking baseline behavior (Stage 2), then to aim at a system objective (in their case, landscape ecological connectivity measured by IIC) subject to mild economic constraints (Stage 3). MIMIC then evolves natural‑language nudges tailored to agent personas (Resistant, Economic, Social) that are evaluated by having a "Farm LLM" edit its baseline code in response to the message (Stage 4). ECHO beats DSPy MiPROv2 on both baseline imitation and global‑target learning, and MIMIC outperforms a DSPy messaging baseline on most farms (Table 1, p.9).

**Strengths:**

This paper presents a genuinely novel synthesis of evolutionary computation and large language models for tackling collective action problems. The framing of ill-structured problems into a series of well-structured subproblems is both conceptually strong and well justified. The implementation of ECHO-MIMIC is technically sound: the evolutionary loop is clear, the operator analysis is detailed, and the authors provide transparent code-fitness tracking. The connection between code complexity and heuristic performance (Fig. 3, p. 7) adds credibility that the model is actually learning something non-trivial rather than just overfitting toy patterns. The integration of behavioral messaging through MIMIC is also compelling, the idea of evolving not just the policy but the persuasive mechanism for its adoption is fresh and well-motivated. Empirically, the method convincingly outperforms a strong baseline (DSPy MiPROv2) on both the imitation and the collective-goal tasks (Table 1, p. 9). Overall, the paper sits at an interesting intersection of AI for social systems, mechanism design, and interpretability research.

**Weaknesses:**

The biggest concern is the narrowness of the experimental setup. All results come from a stylized five-farm synthetic landscape with simplified geometry and an overly discrete target metric (IIC), which makes it hard to judge real-world robustness. Even though ECHO beats the baseline, the absolute accuracies on global-target learning are low (0.13–0.31 across farms, Table 1, p. 9). It’s unclear if the learned heuristics are practically usable or just internally consistent. The use of LLM-simulated “Farm” agents to evaluate nudges is clever but questionable as a proxy for real human behavior, especially since the authors note that outcomes degrade substantially when swapping model families (Gemini 2.0 vs 1.5). Reproducibility across providers and prompts seems fragile. The complexity-performance trade-off (Fig. 3d, p. 7) also undercuts the claim that the method produces “simple, executable heuristics”; the best programs appear relatively opaque and brittle. Finally, some design choices, like mapping directional sets to fractional intervention values or relying on neighbor in-context examples, feel ad-hoc and would likely break under different spatial or economic assumptions.

**Questions:**

1. The framework currently optimizes for the Integral Index of Connectivity (IIC). How dependent are your results on this specific metric?

2. When discussing robustness, it’s unclear whether this refers to stability of learned heuristics under perturbations (e.g., small geometric or parameter changes) or resilience of the optimization loop itself.

---

> ### Author Response · Authors · 2025-11-22
> **Expanded Generalization with New EV Domain, AutoGen Baseline, and Cross-Model Validation**
>
> We thank the reviewer for noticing the strengths in our paper and providing constructive comments. We address your comments in order below.
>
> 1. "The biggest concern is the narrowness of the experimental setup. All results come from a stylized five-farm synthetic landscape with simplified geometry and an overly discrete target metric (IIC), which makes it hard to judge real-world robustness."
>
> We thank the reviewer for this comment. We used IIC because it's a standard metric for ecological connectivity in literature, and maximize it only to arrive at the ground truth for Stage 3 (global outcome). We could as well have used Probability of Connectivity (PC) or Betweenness Centrality (BC) to arrive at different ground truth spatial configurations as a global outcome, and our ECHO-MIMIC pipeline would have tried to achieve that. Therefore, we believe that the choice of the global metric does not affect the real-world robustness of our pipeline. It only helps in defining our desired outcome.
>
> In order to further demonstrate the generalization ability of our pipeline, we have now added a second domain which we have been working on since submission, **Carbon-Aware EV Charging Slot Choice**, with different topology, goals, and metrics. We modify the (state, action) schema, evaluator, observability, topology, and context, and re-use the same ECHO-MIMIC stages. The relative gains over the added baselines persist, supporting domain portability and generalization. A brief introduction to the domain along with the performance is given in the results, and more details about the domain and task is provided in the appendix.
>
> 2. "Even though ECHO beats the baseline, the absolute accuracies on global-target learning are low (0.13–0.31 across farms, Table 1, p. 9)."
>
> Our Stage 3 uses a discrete directional-set score (intersection-over-union on action sets), which is intentionally conservative: small discrepancies count as errors even if they minimally impact the global metric. The point was to prevent “near-misses” from inflating results. You can see from the visual results that the predicted outcomes are very close to the ground truth for the majority of the farms.
>
> 3. "It’s unclear if the learned heuristics are practically usable or just internally consistent."
>
> We agree with the concern. In the current submission we did not enforce an explicit complexity budget nor run a practitioner usability study. Some evolved rules are indeed longer or use thresholds that may feel unintuitive on first read. Our goal in this first paper was to show that an LLM-guided evolutionary process along with a supporting multi-stage pipeline can discover policies that enable collective action; a full audit of “can a typical field agent implement every discovered rule as-is?” is an important next step.
>
> Many of the rules ECHO produces are already in a form that practitioners regularly use. These include, a) directional implementations (e.g., “increase by one level if risk indicator rises; otherwise hold”) are simple, one-line instructions that mirror standard playbooks in resource management and are straightforward to train and monitor. b) thresholded heuristics on readily observed quantities (e.g., treat if expected yield loss < τ and at least one neighbor is treating) are short, “if-then” rules. These are the same structural family as long-standing field heuristics (caps/floors, run-length limits, “stagger with neighbors”), and can be implemented as checklists or decision cards. c) Neighbor-aware staggering (“avoid the most crowded slot among immediate neighbors”, “do not allow more than two untreated parcels in a row”) encodes simple coordination patterns that people already follow informally.
>
> We agree that some learned rules combine several conditions or set thresholds that look odd. Two points help here: a) Complex rules can be operationalized with light scaffolding. Even when a policy is longer than you would hand-write, it can be delivered as a decision aid like a one-page “policy card,” a mobile form with 3-5 inputs and a recommended action, a small lookup table, or a color-coded map overlay. In practice, this translates the program’s conditions into a blueprint the agent can follow. The underlying code remains the same artifact we evolved, but the interface makes it implementable. b) Implementability is context-dependent. A rule that feels “weird” in one setting (e.g., a 0.37 threshold) may be perfectly natural in another once variables are rescaled or discretized (e.g., “≥ 4 units” after conversion). In our pipeline, discretization or rescaling is a data-layer choice, not a change to the learning method.
>
> We deliberately allowed the search to explore beyond hand-written simplicity to understand what the model would discover under performance pressure. This surfaced both very simple patterns (directional, one-threshold, neighbor staggering) and more composite ones. We will prioritize implementability in follow-ups.

---

> ### Author Response · Authors · 2025-11-22
> **Continuing previous comment**
>
> 4. "The use of LLM-simulated “Farm” agents to evaluate nudges is clever but questionable as a proxy for real human behavior, especially since the authors note that outcomes degrade substantially when swapping model families (Gemini 2.0 vs 1.5)."
>
> We agree simulated humans are approximations. We use two safeguards that somewhat help: (a) we measure messages by induced code edits (a concrete behavioral change), and (b) we explicitly enable the choice to refuse any change in behavior.
>
> Further, we now also include the performance of our approach with an *alternate LLM provider/family* in the same capability range (OpenAI GPT-5 nano) and *higher capabilities* (Gemini 2.5 Pro and GPT-5 mini) to quantify drift. Nonetheless, we agree that simulated LLM agents just hint at the right direction, and are not a proxy for real human behaviour. We defer this to future work, to rope in participants in a real world study and deploy our framework in a collective action setup by implementing what we describe in Appendix E “Real-world Applications and Potential Extensions”.
>
> 5. "Reproducibility across providers and prompts seems fragile."
>
> We now include the performance of our approach with an alternate LLM provider/family in the same capability range (OpenAI GPT-5 nano) as our existing model as well as a model pair with higher capabilities (Gemini 2.5 Pro and GPT-5 mini).
>
> We also agree that because we have chosen a specific task to demonstrate our method on, the resulting system instructions and prompts are only applicable to the problem at hand. However, coming up with system instructions and prompts that work across any given collective action problem, is probably not feasible. This is because each problem has its own (state, action) schema, observability constraints, type of ground truth information, background context, additional relevant information, etc, which is only available to the user. This is why our prompt templates are modular and composable, so that given the information available to the user and desired functionality, they can build their own system instructions and prompts.
>
> Nonetheless, we agree that humans writing these instructions can be unreliable and cumbersome, which is why we now have created a **Domain Creation Agent** that automatically writes these instructions in the desired modular format given the (state, action) schema of agents, observability constraints, type of ground truth information, background context, and additional relevant information of the domain specific task at hand. It helps generate system instructions, task prompts, and evaluation harness for the ECHO and MIMIC stages, automatically adapting to the specific terminology and logic of the new domain, allowing our framework to scale to new collective action problems.
>
> We see similar performance irrespective of the LLM family (in the same capability range) and source of instructions, validating that the majority of the gains come from the evolutionary heuristic exploration rather than prompting or choice of model in the same capability range. Further, we verify this with a model pair with higher capabilities (Gemini 2.5 Pro and GPT-5 mini) as well. See Table 1.
>
> 6. "The complexity-performance trade-off (Fig. 3d, p. 7) also undercuts the claim that the method produces “simple, executable heuristics”; the best programs appear relatively opaque and brittle."
>
> Thank you for pointing this out. We agree that we shouldn’t be making the claim that the generated heuristics are “simple”. We now remove any similar claims from the revised version. We agree that there is a tradeoff between complexity and performance, and we acknowledge this in the paper on more than one occasion and also detail follow up experiments or techniques that can potentially mitigate this. We believe that implementing these is beyond the scope of the current paper.
>
> 7. "Finally, some design choices, like mapping directional sets to fractional intervention values or relying on neighbor in-context examples, feel ad-hoc and would likely break under different spatial or economic assumptions."
>
> We agree that mapping directional sets to fractional interventions is ad-hoc and domain/task dependent. In our case, we needed a way to bridge non-directional outputs to directional outputs and that is why we went this route. In a different domain this may not be required, as in the second domain we studied. Our assumption behind this mapping was that if a global coordinator can nudge a local agent to change magnitude (arguably more important) then direction can be according to whatever the global coordinator suggests (as it has global knowledge and knows best).

---

> ### Author Response · Authors · 2025-11-22
> **Continuing previous comment**
>
> We believe that Neighbor ICL is not really ad-hoc. As we argue in the text, Neighbor ICL allows us to test whether the LLM can infer decision rules from analogous contexts when supervision is provided indirectly via EA selection. It also mirrors observational diffusion in communities, where practices propagate through local networks facing shared information. We now demonstrate that this is useful even in our second domain, under different spatial, temporal, and economic context.
>
> Q1. "Framework optimizes for the Integral Index of Connectivity (IIC)"
>
> As discussed above, we maximize IIC only to arrive at the ground truth for Stage 3 (global outcome). We could as well have used Probability of Connectivity (PC) or Betweenness Centrality (BC) to arrive at different ground truth spatial configurations as a global outcome, and our ECHO-MIMIC pipeline would have tried to achieve that. Therefore, we believe that the choice of the global metric does not affect the real-world robustness of our pipeline. It only helps in defining our desired outcome. For more details, see response to your first comment.
>
> Q2. "Robustness unclear"
>
> We mean something closer to the former, including the robustness of the approach across change in topology and also the robustness of the discovered heuristics to dynamic changes in the external environment over time. We now come close to addressing a part of this by testing on a second domain of Carbon-Aware EV Charging Slot Choice, with different topology, goals, and metrics. We consider that addressing the robustness to dynamic changes in the external environment is future work.
>
> Apart from our discussion here, we added an additional baseline (AutoGen), comparison with stronger base LLMs (Gemini 2.5 Pro and GPT-5 mini), a second domain (Carbon-Aware EV Charging Slot Choice), a Domain Creation Agent, and an additional LLM family (GPT-5 nano). We also demonstrated generalization, expanded the nudging section and literature review, and clarified the core idea and novelty. We hope that these changes have made our paper stronger than the previous version.

---

### Official Review · Reviewer_KLsp · 2025-10-31

**Soundness:** 2
**Presentation:** 1
**Contribution:** 2
**Rating:** 2
**Confidence:** 3

**Summary:**

This work presents ECHO-MIMIC, an evolutionary algorithm-driven LLM framework that converts ill-structured problems into effective, well-structured problems. It leverages LLM optimizers with evolutionary algorithms to optimize code scripts per agent for optimizing their individual behaviors. The framework is studied in a simulation of an agricultural landscape and compared against a single prompt optimization baseline, where ECHO-MIMIC shows performance improvements in accuracy, discovering high-performing heuristics for individual agents.

**Strengths:**

1. This work studied a novel problem in agricultural landscape management, reflecting real-world impacts on resource management problems.

2. The improvements over the MIPRO baseline are large, validating the effectiveness of the proposed framework against general prompt optimization approaches.

3. Using the code representation for controlling individual behavior provides a good optimization space given the current capability of large language model optimizers.

**Weaknesses:**

1. Though the agricultural landscape management provides a novel context for studying the ECHO-MIMIC approach, the research is also strongly limited to the domain of farm management. The system instructions and the optimizers are highly overfitted to this particular type of problem. Meanwhile, the human engineering of prompts can make the system brittle when the backbone LLM is transferred to another family of backbone. Therefore, the generalization of the proposed approach to other domains is understudied.

 2. The evolutionary algorithm part within the ECHO-MIMIC framework is not fundamentally different compared to earlier LLM-EA approaches (e.g., EvoPrompt), which limits the technical contribution of this work.

3. Only a single baseline, MIPROv2, has been studied in this work. There are many other automatic multi-agent optimization frameworks that are worthy of comparison, such as AutoGen and G-Designer [1].

[1] Zhang, Guibin, et al. "G-designer: Architecting multi-agent communication topologies via graph neural networks." arXiv preprint arXiv:2410.11782 (2024).

**Questions:**

1. How do you extend the proposed approach to other domains besides the agricultural collective action studied in this work?

---

> ### Author Response · Authors · 2025-11-22
> **Added EV Domain, AutoGen Baseline, and Cross-Model Validation**
>
> We thank the reviewer for noticing the strengths in our paper and providing constructive comments. We address your comments in order below.
>
> 1. "Generalization of the proposed approach to other domains is understudied"
>
> We thank the reviewer for this comment. We want to clarify that our proposed ECHO-MIMIC pipeline is not tailored to agriculture; We are just demonstrating its effectiveness by taking agriculture as an example domain. The pipeline remains the same when we change (a) the agent state schema, (b) the allowable actions, and (c) the local and global evaluators.
>
> We agree that because we chose a specific task to demonstrate our method on, the resulting system instructions and prompts are only applicable to the problem at hand. However, coming up with system instructions and prompts that work across any given collective action problem, is probably not feasible. This is because each problem has its own (state, action) schema, observability constraints, type of ground truth information, background context, additional relevant information, etc, which is only available to the user. This is why our prompt templates are modular and composable, so that given the information available to the user and desired functionality, they can build their own system instructions and prompts.
>
> Nonetheless, we agree that humans writing these instructions can be unreliable and cumbersome, which is why we now have created a **Domain Creation Agent** that automatically writes these instructions in the desired modular format given the (state, action) schema of agents, observability constraints, type of ground truth information, background context, and additional relevant information of the domain specific task at hand. It helps generate system instructions, task prompts, and evaluation
> harness for the ECHO and MIMIC stages, automatically adapting to the specific terminology and logic of the new
> domain, allowing our framework to scale to new collective action problems.
>
> Moreover, to test the brittleness of both our human generated instructions and the agent generated instructions, we noted the performance of our pipeline while using an alternate LLM from OpenAI (GPT-5 nano) that is at a somewhat similar level of capability as our original LLM (gemini-2.0-flash-thinking-exp-01-21). We see similar performance irrespective of the LLM family and source of instructions, validating that the majority of the gains come from the evolutionary heuristic exploration rather than prompting or choice of model in the same capability range. Further, we verify this with a *model pair with higher capabilities* (Gemini 2.5 Pro and GPT-5 mini) as well.  See Table 1.
>
> We agree that we didn’t do sufficient justice in showing generalization to more than one domain. We had already started working on demonstrating our method on additional domains after submission. We have now added a *second domain*, **Carbon-Aware EV Charging Slot Choice**, with different topology and goals. We modify the (state, action) schema, evaluator, observability, topology, and context, and re-use the same ECHO-MIMIC stages. The relative gains over the added baselines persist, supporting domain portability. A brief introduction to the domain along with the performance is given in the results, and more details about the domain and task is provided in the appendix.
>
> We elaborate more in the paper about how given the required information about the task, our Domain Creation Agent, modular and composable instruction format, and the ECHO-MIMIC pipeline overall, our approach generalizes across domains as demonstrated. We believe this makes ECHO-MIMIC a powerful way to think about collective action problems, and a novel and important use of LLMs under the field of AI for public/social good.
>
> 2. "Not different than LLM-EA approaches like EvoPrompt"
>
> Our contribution is not *EA over prompts*; it is and end-to-end framework to think about and drive collective action. it is EA over executable policies with a compile/repair loop, neighbor-aware in-context examples, and a downstream messaging module measured by code edits. It is observing that the ill-structured problem of collective action can be broken down into well-structured problems and LLMs can be used creatively to enable this. That is substantially different from evolving free-text prompts. Our argument is that this whole pipeline results in a powerful mechanism that enables collective action. Therefore, we disagree that the technical contribution of this work is limited, as the problem formulation and breakdown are non-trivial. This is supported by the fact that there exists no other work in this direction. We now clarify this perspective at multiple places in the paper.

---

> ### Author Response · Authors · 2025-11-22
> **Continuing previous comment**
>
> 3. "Only a single baseline"
>
> We agree that comparing our work with only one baseline MIPROv2 is not sufficient. Thank you for pointing us to relevant prior work. We now discuss these two methods a bit in related work. We now also include comparisons with **AutoGen** in our new version. For AutoGen, we implement a multi-agent workflow consisting of a Planner, a Producer, and a Critic. The Planner first breaks down the problem into concrete steps. Then, the Producer generates the required artifact, either a Python script for policy synthesis or a natural language message for nudging. Finally, a Critic reviews the output for compliance with input/output constraints and logical correctness, triggering a revision loop if necessary before execution.
>
> We believe G-Designer is related but orthogonal to our work as it learns the communication topology for multi-agent LLM systems. Our setup fixes the neighbor graph and focuses on program (policy) synthesis + measurable nudging, not on learning who talks to whom. Therefore, we omit this comparison. See Table 1.
>
> Q1. "How do you extend to other domains"
>
> Thank you for the question. We have already covered this in detail in Appendix E, “Real-world Applications and Potential Extensions”. In the revised version, we now also demonstrate extension by adding a second domain of EV Charging and creating a Domain Creation Agent for automatically plugging in the user's task into our workflow .
>
> Our Domain Creation Agent agent takes as input a high-level domain schema of: a) Agent State: Description of local variables (e.g., crop yields, charging demand). b) Action Space: Allowable decisions (e.g., intervention length, slot usage). c) Observability: What neighbors or global signals are visible. d) Constraints: Budget, physical limits, or regulatory bounds. Using a meta-prompt, the agent generates the specific system instructions, task prompts, and evaluation harness for the ECHO and MIMIC stages. This ensures that the prompt and evaluation templates are modular and composable, automatically adapting to the specific terminology and logic of the new domain, allowing our framework to scale to new collective action problems. We talk about this aspect in the paper.
>
> In summary, we added an additional baseline (AutoGen), a second domain (Carbon-Aware EV Charging Slot Choice), an additional LLM family (GPT-5 nano), a Domain Creation Agent, and demonstrated generalization and clarified novelty. Apart from our discussion, we also added a comparison with stronger base LLMs (Gemini 2.5 Pro and GPT-5 mini), expanded the nudging section and literature review, improved  the figures, and clarified the core idea. We hope that these changes have made our paper much stronger than the previous version, and we respectfully ask you to reconsider your rating for our paper.

---

### Author Response · Authors · 2025-11-22
**Cover Note**

We thank the reviewers for thoughtful and constructive feedback. We proposed ECHO-MIMIC, an end-to-end framework to think about collective action, where we used the strengths of the LLM+EA paradigm to discover implementable heuristics for agent behaviour and persuasive messages to nudge agents towards globally important outcomes. This turns an ill-structured collective-action problem into a small set of well-structured sub-problems per agent.

Below we summarize what we changed and how these changes directly address the concerns raised across reviews.

1. Added a second domain of **Carbon-Aware EV Charging Slot Choice** with different topology, objectives, and evaluator; we reuse the identical ECHO-MIMIC pipeline. The relative gains persist, supporting portability beyond agriculture and generalization across domains.

2. Created **Domain Creation Agent** that automatically generates system instructions, task prompts, and evaluation harness for the ECHO and MIMIC stages for any domain, automatically adapting to the specific terminology and logic of the new domain, allowing our framework to scale to new collective action problems.

3. Added an additional baseline, a multi-agent LLM scaffold (**AutoGen**) under identical observability, to test whether a generic agent team can match our pipeline. The results show demonstrate our superior performance versus baselines across domains.

4. Compared with *additional LLM families*, both with a comparable-capability OpenAI GPT-5 nano (vs. our original Gemini 2.0 Flash Thinking variant), and a stronger pair (Gemini 2.5 Pro and GPT-5 mini) to show scaling trends and family sensitivity.

Clarity & presentation

1. Clarified the core idea of the work and novelty, i.e., a general end-to-end framework to drive collective action.
2. Expanded nudging section (who nudges, what information is used, explicit refusal).
3. Added a related-work subsection on AI and social dilemmas and how ECHO-MIMIC differs (applied spatial setting; executable heuristics; end-to-end).
4. Reworked and reorganized diagrams. Removed unnecessary pieces of text.
5. Removed blanket claim that discovered heuristics are always simple.

Across reviews we heard three themes: generality, baselines, and practical clarity. We responded by adding a second domain which we were already working on, including a strong agent-scaffold baseline, checking model-family robustness, expanding nudging details, and improving presentation. We hope these changes address the concerns raised by some reviewers and reinforce the positive assessments by others. We respectfully ask the AC and reviewers to reconsider the paper in light of these additions and clarifications.

---

### Meta-Review · Area_Chair_ie2G · 2026-01-07

**Summary:**

The paper proposes ECHO-MIMIC, a framework combining Large Language Models (LLMs) and Evolutionary Algorithms (EA) to address collective action problems. The approach decomposes Ill-Structured Problems into tractable heuristics (ECHO) and persuasive natural language messages (MIMIC). The authors demonstrated the framework initially on an agricultural landscape management task and, during the rebuttal, extended it to an EV charging domain.

While the paper presents a novel synthesis of LLM-based program search and mechanism design, there are two primary factors driving this decision:

1.  **Policy Violation (Anonymity):** Reviewer KLsp identified a significant violation of the ICLR double-blind policy. The supplementary material (specifically log files) contains file paths revealing the author's identity ("Kevin..."). Under conference rules, this typically constitutes grounds for rejection regardless of technical merit, as it compromises the integrity of the review process.
2.  **Validation of Behavioral Components:** While the authors added a second domain (EV charging) and baselines (AutoGen) in response to feedback, a fundamental concern raised by Reviewer mHoj and alluded to by others remains: the framework relies entirely on LLM-simulated agents to validate the "nudging" mechanism. The effectiveness of the MIMIC module assumes that real humans would respond to generated messages similarly to an "Agent LLM" with a persona. This sim-to-real gap in behavioral modeling is substantial, and without human-subject validation or stronger theoretical grounding in behavioral economics beyond LLM proxies, the claims regarding "persuasive rationales" and real-world applicability are difficult to substantiate for a top-tier venue.

**Reviewer Concerns:**

The presence of author names in the code logs (Reviewer KLsp) is an unresolved policy violation. This paper should be desk-rejected.

**Reviewer Scores:**

This paper should be desk-rejected according to the double-blind policy.

---

### Decision · Program_Chairs · 2026-01-26

Reject